# Multitask Learning with Stochastic Interpolants

**Hugo Negrel**
Capital Fund Management
23 Rue de l'Université, 75007 Paris
`hugonegrel13@gmail.fr`

**Florentin Coeurdoux**
Capital Fund Management
23 Rue de l'Université, 75007 Paris
`florentin.coeurdoux@cfm.com`

**Michael S Albergo**
Society of Fellows, Harvard University
`malbergo@fas.harvard.edu`

**Eric Vanden-Eijnden**
Machine Learning Lab
Capital Fund Management
23 Rue de l'Université, 75007 Paris
`eric.vanden-eijnden@cfm.com` [*]

## Abstract

We propose a framework for learning maps between probability distributions that broadly generalizes the time dynamics of flow and diffusion models. To enable this, we generalize stochastic interpolants by replacing the scalar time variable with vectors, matrices, or linear operators, allowing us to bridge probability distributions across multiple dimensional spaces. This approach enables the construction of versatile generative models capable of fulfilling multiple tasks without task-specific training. Our operator-based interpolants not only provide a unifying theoretical perspective for existing generative models but also extend their capabilities. Through numerical experiments, we demonstrate the zero-shot efficacy of our method on conditional generation and inpainting, fine-tuning and posterior sampling, and multiscale modeling, suggesting its potential as a generic task-agnostic alternative to specialized models.

## 1 Introduction

Recent years have witnessed remarkable advances in generative modeling, with transport-based approaches such as normalizing flows (Lipman et al., 2022; Albergo and Vanden-Eijnden, 2022; Liu et al., 2022) and diffusion models (Ho et al., 2020; Song and Ermon, 2020; De Bortoli et al., 2021; Albergo et al., 2023a) emerging as state-of-the-art techniques across various application domains (Rombach et al., 2022; Mazé and Ahmed, 2023; Alverson et al., 2024). These methods have revolutionized our ability to generate high-quality images, text, and other complex data types by viewing these data as samples from an unknown target distribution and learning to transform simple (e.g., noise) distributions into this target. This transformation is effectively achieved via transport of the samples by a flow or diffusion process with a drift (or score) parameterized by neural networks and estimated via simulation-free quadratic regression, enabling highly efficient training.

Despite their impressive performance, these generative frameworks face a fundamental limitation: they are typically designed and trained for specific, predetermined tasks, with the generative objective specified before training. For example, a diffusion model trained to generate images cannot easily be repurposed to perform inpainting or other editing tasks without substantial modification or retraining.

---

[*]also at: Courant Institute of Mathematical Sciences, New York University, New York, NY 10012, USA, `eve2@cims.nyu.edu`

39th Conference on Neural Information Processing Systems (NeurIPS 2025).

While some flexibility can be achieved through conditioning variables or prompting, these approaches remain constrained within narrowly defined operational boundaries established beforehand. Recent attempts at multitask generation using approximated guidance strategies (Chung et al., 2023; Song et al., 2022; Wang et al., 2024) have shown promise, but rely on uncontrolled approximations that limit their theoretical guarantees and can lead to unpredictable results. These methods typically operate within a predefined space of capabilities and lack the flexibility to adapt to novel tasks without retraining, often requiring domain-specific architectural modifications or specialized training procedures that further limit their versatility.

In this paper, we introduce a novel framework for training truly multi-task generative models based on a generalized formulation of stochastic interpolants. Our key insight is to replace the scalar time variable traditionally used in transport-based models with a linear operator. These *operator-based interpolants* enable interpolation between random variables across multiple dimensional planes or setups, providing a unified mathematical formulation that treats various generative tasks as different ways of traversing the same underlying space, rather than as separate problems requiring distinct models. This dramatically expands the space of possible tasks that a single model can perform.

Our **main contributions** include theoretical advances that establish a framework for multiple generative applications:

- We extend traditional scalar interpolation in dynamical generative models to higher-dimensional structures, developing a unified mathematical formulation of ***operator-based interpolants*** that treats various generative tasks as different ways of traversing the same underlying space. This opens up fundamentally new ways of seeing how generative models can be structured to handle multiple objectives simultaneously.

- We show how this framework enables generative models with ***continual self-supervision*** over a wide purview of generative tasks, making possible: **universal inpainting models** that work with arbitrary masks, **multichannel data denoisers** with operators in the Fourier domain, **posterior sampling** with quadratic rewards, and **test-time dynamical optimization** with rewards and interactive user feedback, all with one pretrained model.

- We demonstrate these various tools on image-infilling, data de-corruption, statistical physics simulation, and dynamical robotic planning tasks across a number of datasets, showing that our method matches or surpasses existing approaches without being specifically tied to any single generative objective.

All common augmentations like text-conditioning and guidance can still be used just as before in our setup. While our approach increases the complexity of the initial learning problem, we provide arguments that this additional complexity can be addressed through scale. In essence, the "pretraining" phase becomes more challenging, but the resulting model gains substantial flexibility and versatility that compensates for the pretraining costs. That is, our approach can be seen as *a way of amortizing learning over a variety of tasks.* This points toward a more general paradigm of universal generative models that can be trained once and then applied to a variety of objectives, potentially reducing computational and environmental costs associated with training separate models for each task.

## 1.1 Related works

**Flow matching and diffusion models.** Our approach extends the theoretical groundwork established in flow matching with stochastic interpolant and rectified flows (Lipman et al., 2022; Albergo and Vanden-Eijnden, 2022; Albergo et al., 2023a; Liu et al., 2022) as well as the probability flow formulations in diffusion models (Ho et al., 2020; Song et al., 2020). which established techniques for handling multiple target distributions simultaneously. Unlike data-dependent coupling approaches (Albergo et al., 2023b) that require constructing specific couplings for tasks like inpainting, our method learns a general operator space that naturally accommodates such tasks without additional coupling design. By introducing operator-valued interpolants, we enable a richer space of transformations between distributions, unlocking a flexible framework for multiple generative tasks.

**Inverse problems and inpainting** Our framework offers a unified approach to inverse problems, contrasting with traditional methods that require problem-specific variational optimization procedures Pereyra et al. (2015). Recent diffusion-based approaches Chung et al. (2023); Song et al. (2022); Kawar et al. (2022); Martin et al. (2025) typically need guided sampling trajectories tailored to

each task. Similarly, methods using MCMC/SMC sampling Coeurdoux et al. (2024); Sun et al. (2024), variational approximations Mardani et al. (2023); Alkan et al. (2023), or optimization-driven techniques Wang et al. (2024); Daras et al. (2022) remain fundamentally task-specific. Our approach encodes solution paths within the interpolant operator structure itself, enabling multiple inverse problems to be addressed through appropriate operator path selection during inference—without additional training. Our work can also be seen as a way to formalize methods that seek to clean data corrupted in various ways Bansal et al. (2022)

**Multiscale and any-order generation** Recent approaches to generative modeling have explored hierarchical strategies through progressive refinement. Visual Autoregressive Modeling (Tian et al., 2024) employs next-scale prediction, while Fractal Generative Models (Li et al., 2025) utilize self-similar structures for multiscale representations. These methods typically constrain generation to fixed paths established during training. In contrast, our framework decouples the training process from generation trajectories, allowing flexible path selection at inference time. This is conceptually related to optimizing generation order in discrete diffusion (Shi et al., 2025) and token ordering studies (Kim et al., 2025), but provides greater flexibility by enabling post-training optimization and dynamic, self-guided generation strategies that adapt based on intermediate results.

## 2 Theoretical framework

Imagine we want to create a single generative model capable of multiple tasks - sampling new data, inpainting, denoising, and more. To achieve this, we need to expand beyond the traditional "single path" between noise and data. In this section, we develop the theoretical foundations of operator-based interpolants, which allow flexible navigation through a richer multidimensional space and will enable multitask generative capabilities discussed in Section 3.

### 2.1 Operator-based interpolants

Suppose that we are given a couple of random variables $(x_0, x_1)$ both taking values in a Hilbert space $\mathcal{H}$ (for example $\mathbb{R}^d$) and drawn from a joint distribution $\mu(dx_0, dx_1)$. Our aim is to design a transport between a broad class of distributions supported on $\mathcal{H}$ involving *mixtures of $x_0$ and $x_1$*. We will do so by generalizing the framework of stochastic interpolant.

**Definition 2.1** (Operator-based interpolants). *Let $B(\mathcal{H})$ be a connected set of bounded linear operators on $\mathcal{H}$ and let $S \subseteq B(\mathcal{H}) \times B(\mathcal{H})$, also connected. Given any pair of linear operators $(\alpha, \beta) \in S$, the* operator-based interpolant $I(\alpha, \beta)$ *is the stochastic process given by*

$$I(\alpha, \beta) = \alpha x_0 + \beta x_1, \qquad (\alpha, \beta) \in S, \tag{1}$$

*where $(x_0, x_1) \sim \mu$. We will denote by $\mu_{\alpha,\beta}$ the probability distribution of $I(\alpha, \beta)$.*

If we picked e.g. $\alpha = (1 - t)\,\mathrm{Id}$ and $\beta = t\,\mathrm{Id}$ with $t \in [0, 1]$, respectively, we would go back to a standard stochastic interpolant, but we stress that the operator-based interpolants from Definition 2.1 are much more general objects. For example, if $\mathcal{H} = \mathbb{R}^d$, we could take $B(\mathcal{H})$ to be a set of $d \times d$ matrices with real entries – choices of $\alpha$, $\beta$ tailored to several multitask generation will be discussed in Section 3. *The main objective of this paper is to show how to exploit this flexibility of design.* Specifically we will show that we can learn a model that can be used to transport samples of $I(\alpha, \beta)$ along *any* paths of $(\alpha, \beta)$ in $S$ without having to choose any such path during training. This will mean that transport problems in a broad class associated to a variety of tasks, including inpainting and block generation of any order, fine-tuning, etc. will be *pretrained* into this model.

### 2.2 Multipurpose drifts and score

To proceed, we introduce two drifts, which are functions of $S \times \mathcal{H}$ taking value in $\mathcal{H}$:

**Definition 2.2** (Multipurpose drift). *The drifts $\eta_0, \eta_1 : S \times \mathcal{H} \to \mathcal{H}$ are given by*

$$\eta_0(\alpha, \beta, x) = \mathbb{E}[x_0 | I(\alpha, \beta) = x], \qquad \eta_1(\alpha, \beta, x) = \mathbb{E}[x_1 | I(\alpha, \beta) = x], \tag{2}$$

*where $\mathbb{E}[ \ \cdot \ |I(\alpha, \beta) = x]$ denotes expectation over the coupling $(x_0, x_1) \sim \mu$ conditioned on the event $I(\alpha, \beta) = x$.*

Using the $L^2$ characterization of the conditional expectation, these drifts can be estimated via solution of a tractable optimization problem with an objective function involving an expectation:

**Lemma 2.3** (Drift objective). *Let $\nu(d\alpha, d\beta)$ be a probability distribution whose support is $S$. Then the drifts $\eta_{0,1}(\alpha, \beta, x)$ in Definition 2.2 can be characterized globally for all $(\alpha, \beta) \in S$ and all $x \in \operatorname{supp}(\mu_{\alpha, \beta})$ via solution of the optimization problems*

$$\eta_0 = \underset{\hat{\eta}_0}{\arg\min} \, \mathbb{E}_{\substack{(\alpha, \beta) \sim \nu \\ (x_0, x_1) \sim \mu}} \left[ \|\hat{\eta}_0(\alpha, \beta, I(\alpha, \beta)) - x_0\|^2 \right], \tag{3}$$

$$\eta_1 = \underset{\hat{\eta}_1}{\arg\min} \, \mathbb{E}_{\substack{(\alpha, \beta) \sim \nu \\ (x_0, x_1) \sim \mu}} \left[ \|\hat{\eta}_1(\alpha, \beta, I(\alpha, \beta)) - x_1\|^2 \right], \tag{4}$$

*where $\| \cdot \|$ denotes the norm in $\mathcal{H}$.*

This lemma is proven in Appendix A. Below we will use (3) and (4) to learn $\eta_{0,1}$ over a rich parametric class made of deep neural networks. Note that the drifts $\eta_0$ and $\eta_1$ are not linearly independent since the definition of $I(\alpha, \beta)$ in (9) together with the equality $x = E[I(\alpha, \beta)|I(\alpha, \beta) = x]$ imply that

$$x = \alpha\eta_0(\alpha, \beta, x) + \beta\eta_1(\alpha, \beta, x). \tag{5}$$

Therefore we can obtain $\eta_0$ from $\eta_1$ if $\alpha$ is invertible and $\eta_1$ from $\eta_0$ if $\beta$ is invertible. Note also that, when $x_0$ is Gaussian and $x_0 \perp x_1$, $\eta_0$ is related to the score of the distribution of the stochastic interpolant:

**Lemma 2.4** (Score). *Assume that $\mathcal{H} = \mathbb{R}^d$ and that the probability distribution $\mu_{\alpha, \beta}$ of the stochastic interpolant $I(\alpha, \beta)$ is absolutely continuous with respect to the Lebesgue measure with density $\rho_{\alpha, \beta}(x)$. Assume also that $x_0 \sim N(0, \operatorname{Id})$ and $x_0 \perp x_1$. Then the score $s_{\alpha, \beta}(x) = \nabla \log \rho_{\alpha, \beta}(x)$ is related to the drift $\eta_0(\alpha, \beta, x)$ via*

$$\eta_0(\alpha, \beta, x) = -\alpha s_{\alpha, \beta}(x). \tag{6}$$

This lemma is proven in Appendix A.

## 2.3 Transport with flows and diffusions

We can now state the main theoretical results of this paper: if we are able to sample the stochastic interpolant $I(\alpha_0, \beta_0)$ at a specific value $(\alpha_0, \beta_0) \in S$, then we can produce sample of $I(\alpha_t, \beta_t)$ along *any* curve $(\alpha_t, \beta_t) \in S$ by solving either a probability flow ODE or an SDE, assuming we have estimated the drifts $\eta_{0,1}(\alpha, \beta, x)$ from Definition 2.2 along this curve:

**Proposition 2.5** (Probability flow). *Let $(\alpha_t, \beta_t)_{t \in [0,1]}$ be any one-parameter family of operators $(\alpha_t, \beta_t) \in S$. Assume that $\alpha_t, \beta_t$ are differentiable for all $t \in [0, 1]$. Then, for all $t \in [0, 1]$, the law of $I(\alpha_t, \beta_t)$ is the same as the law of the solution $X_t$ to*

$$\dot{X}_t = \dot{\alpha}_t \eta_0(\alpha_t, \beta_t, X_t) + \dot{\beta}_t \eta_1(\alpha_t, \beta_t, X_t), \qquad X_0 \overset{d}{=} I(\alpha_0, \beta_0). \tag{7}$$

This proposition is proven in Appendix A. Similarly, for generation with an SDE, we have:

**Proposition 2.6** (Diffusion). *Assume that $\mathcal{H} = \mathbb{R}^d$ and the probability distribution $\mu_{\alpha, \beta}$ of the stochastic interpolant $I(\alpha, \beta)$ is absolutely continuous with respect to the Lebesgue measure. Assume also that, in $I(\alpha, \beta)$, $x_0$ is Gaussian and $x_0 \perp x_1$. Then, under the same conditions as in Proposition 2.5, if $\alpha_t$ is invertible, for all $t \in [0, 1]$ and any $\epsilon_t \geqslant 0$, the law of $I(\alpha_t, \beta_t)$ is the same as the law of the solution $X_t^\epsilon$ to*

$$dX_t^\epsilon = \left( \dot{\alpha}_t - \epsilon_t \alpha_t^{-1} \right) \eta_0(\alpha_t, \beta_t, X_t^\epsilon) dt + \dot{\beta}_t \eta_1(\alpha_t, \beta_t, X_t^\epsilon) dt + \sqrt{2\epsilon_t} dW_t, \quad X_0^\epsilon \overset{d}{=} I(\alpha_0, \beta_0). \tag{8}$$

*where $W_t$ is a Wiener process in $\mathbb{R}^d$.*

This proposition is also proven in Appendix A. Note that the SDE (8) reduces to the ODE (7) if we set $\epsilon_t = 0$. Note that if $\alpha_t$ is positive-definite we can use $\bar{\epsilon}_t = \epsilon_t \alpha_t^{-1}$ as new diffusion coefficient, which set the noise term in (8) to $\sqrt{2\bar{\epsilon}_t} \alpha_t^{1/2} dW_t$; this allows to extend this SDE to paths along which we can have $\alpha_t = 0$.

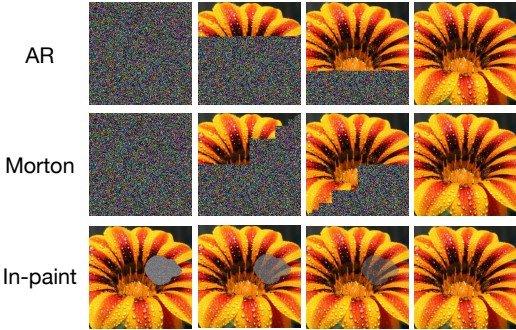

**Figure 1: Multi-task, self-supervised sampling:** Schematic representation of various sub-tasks that are captured by the minimizer of our learning objective using the Hadamard-product interpolant in (9). A generative task is chosen in a zero-shot manner by specifying $\alpha$ as a function of time after training. This $\alpha_t$ serves as a continual self-supervision of what has been unmasked vs. what remains. **Top**: $\alpha_t$ is chosen to generate pixels in an autoregressive fashion. **Middle**: $\alpha_t$ is chosen to sample along a fractal morton order. **Bottom**: $\alpha_t$ can be chosen to do zero-shot inpainting.

## 3  Multitask generation

In this section we discuss how to use the theoretical framework introduced in Section 2 to perform multiple generative tasks without having to doing any retraining.

### 3.1  Self-supervised generation and inpainting

Inpainting—the task of filling in missing parts of an image—traditionally requires specialized training for each possible mask configuration. Our operator-based framework enables a fundamentally different approach: a single model that can perform inpainting with any arbitrary mask, chosen at inference time, or can generate samples from scratch in an arbitrary ordering of the generation. This includes standard generation of all dimensions at once, autoregressive generation dimension by dimension, blockwise fractal generation, and so forth. In particular, we may want to construct a generative model that fills in missing entries from a sample $x_1 \in \mathbb{R}^d$ drawn from a data distribution $\mu_1$. We would like this model to be universal, in the sense that it can be used regardless of which entries are missing; their number and position can be arbitrary and changed post-training, allowing for flexible inpainting and editing (see Figure 1 for an illustration). This approach creates a natural self-supervision mechanism, as the model continuously tracks which parts have been generated and which remain to be filled.

To perform this task, assume that $x_1$ is drawn from the data distribution $\mu_1$ of interest and $x_0$ drawn independently from $N(0, \mathrm{Id})$, so that $\mu = N(0, \mathrm{Id}) \times \mu_1$, set $\beta = 1 - \alpha$ in the operator interpolant (1), and assume that $\alpha$ is a diagonal matrix. With a slight abuse of notations we can then identify the diagonal elements of the matrix $\alpha$ with a vector $\alpha \in \mathbb{R}^d$ and write (1) as

$$I(\alpha) = \alpha \odot x_0 + (1 - \alpha) \odot x_1, \tag{9}$$

where $\odot$ denotes the Hadamard (i.e. entrywise) product. The drift to learn in this case is

$$\eta(\alpha, x) = \mathbb{E}\big[x_0 - x_1 | \alpha \odot x_0 + (1 - \alpha) \odot x_1 = x\big] \tag{10}$$

for $\alpha \in [0, 1]^d$, and this learning can be done via solution of

$$\min_{\hat\eta} \mathbb{E}_{\substack{\alpha \sim U([0,1]^d) \\ x_0 \sim N(0,\mathrm{Id}) \\ x_1 \sim \mu_1}} \big[\|\hat\eta(\alpha, I(\alpha \odot x_0 + (1 - \alpha) \odot x_1) + x_1 - x_0)\|^2\big], \tag{11}$$

Denoting $x_1 = (x_1^1, x_1^2, \ldots, x_1^d)$, suppose that we observe $x_1^i$ for the entries with $i \in \sigma \subset \{1, \ldots, d\}$ and would like to infer the missing entries with $i \in \sigma^c = \{1, \ldots, d\} \setminus \sigma$. To perform this inpainting we can use the probability flow ODE (7) with a path $\alpha_t$ such that $\alpha_t^i = 0$ if $i \in \sigma$ and $\alpha_t^i = 1 - t$ if $i \in \sigma^c$. Note that this can be done for *any choice of $\sigma$ without retraining*.

### 3.2  Multichannel denoising

Suppose that $B_1, B_2, \ldots B_n$ are deterministic corruption operators that can be applied to the data. For example $B_1$ could be a high-pass filter, $B_2$ a motion blur, etc. and they could be defined primarily in the Fourier representation of the data. Similarly, if $x_0 \sim N(0, \mathrm{Id})$, let $A_1, A_2, \ldots A_m$ be operators



Gaussian

Motion

**Figure 2: Multichannel denoising:** Possible interpolations fulfilled by various choices of operators in (12). We present two such examples in the form of Gaussian and motion blurring, realized by interpolations defined in the Fourier domain.

that give some structure to this noise (e.g. some spatial correlation over the domain of the data). Set $A_0 = B_0 = \mathrm{Id}$ and take

$$\alpha = \sum_{i=0}^{m} a_i A_i, \qquad \beta = \sum_{i=0}^{n} b_i B_i \tag{12}$$

where $a_i, b_i$ are nonegative scalars each taking values in some range that includes 0 and $b_0 = 1$. With this choice we can find path $(\alpha_t, \beta_t)_{t \in [0,1]}$ that bridges data $x_1 \sim \mu$ corrupted in any possible channel as $\sum_{i=1}^{m} a_i A_i x_0 + \sum_{i=1}^{n} b_i B_i x_1$ for some choice of $(a_1, a_2, \ldots, b_1, b_2, \ldots)$ back to the clean data via a path that bridges this choice of parameters to $b_0 = 1$ and $a_0 = a_1 \ldots = b_1 = \ldots = 0$. See Figure 2 for an illustration of two possible corruption schemes.

### 3.3 Fine-tuning and posterior sampling

Suppose that we are given data from a distribution $\mu_1(dx)$, and that we would like to generate samples from $\mu_1^r(dx) = Z^{-1} e^{r(x)} \mu_1(dx)$ where $r : \mathcal{H} \to \mathbb{R}$ is a reward function and $Z = \int_{\mathcal{H}} e^{r(x)} \mu_1(dx)$ is a normalization function, which is unknown to us but we assume finite – in the context of Bayesian inference, $\mu_1$ plays the role of prior distribution, $r$ is the likelihood, and $\mu_1^r$ is the posterior distribution. We will assume that the reward $r$ is a quadratic function, i.e.

$$r(x) = \tfrac{1}{2}\langle x, Ax \rangle + \langle b, x \rangle \tag{13}$$

where $A$ is a definite negative bilinear operator on $\mathcal{H}$, $b \in \mathcal{H}$, and $\langle \cdot, \cdot \rangle$ denotes the inner product on $\mathcal{H}$. For simplicity we also assume that $\mathcal{H} = \mathbb{R}^d$: the general case can be treated similarly.

In this context, assume that we have learned the drifts $\eta_{0,1}$ associated with the interpolant

$$I(\alpha, \beta) = \alpha x_0 + \beta x_1, \qquad x_0 \sim N(0, \mathrm{Id}), \quad x_1 \sim \mu_1, \quad x_0 \perp x_1 \tag{14}$$

so that we can generate samples from the prior distribution. Our next result shows that this gives us access to the drifts $\eta_{0,1}^r$ associated with the interpolant

$$I_r(\alpha, \beta) = \alpha x_0 + \beta x_1^r, \qquad x_0 \sim N(0, \mathrm{Id}), \quad x_1^r \sim \mu_1^r, \quad x_0 \perp x_1^r \tag{15}$$

involving data $x_1^r$ from the posterior distribution.

**Proposition 3.1.** *Let*

$$\eta_0(\alpha, \beta, x) = \mathbb{E}[x_0 | I(\alpha, \beta) = x], \qquad \eta_1(\alpha, \beta, x) = \mathbb{E}[x_1 | I(\alpha, \beta) = x], \tag{16}$$

*be the drifts associated with the interpolant* (14) *and*

$$\eta_0^r(\alpha, \beta, x) = \mathbb{E}[x_0 | I_r(\alpha, \beta) = x], \qquad \eta_1^r(\alpha, \beta, x) = \mathbb{E}[x_1^r | I_r(\alpha, \beta) = x], \tag{17}$$

*be the drifts associated with the interpolant* (15). *If $\alpha$ and $\beta$ are invertible, then*

$$\eta_0^r(\alpha, \beta, x) = \alpha^{-1}\beta\beta_r^{-1}\alpha_r \eta_0(\alpha_r, \beta_r, x_r) + \alpha^{-1}(x - \beta\beta_r^{-1}x_r) \tag{18}$$

$$\eta_1^r(\alpha, \beta, x) = \eta_1(\alpha_r, \beta_r, x_r) \tag{19}$$

*as long as we can find a pair $(\alpha_r, \beta_r)$ that satisfies*

$$\beta_r^T \alpha_r^{-T} \alpha_r^{-1} \beta_r = \beta^T \alpha^{-T} \alpha^{-1} \beta - A \tag{20}$$

*and $x_r$ is given by*

$$x_r = \alpha_r \alpha_r^T \beta_r^{-T} \left( \beta^T \alpha^{-T} \alpha^{-1} x + b \right). \tag{21}$$

This proposition is proven in Appendix A as a corollary of Proposition A.1 that relates the probability distribution of $I_r$ to that of $I$. Proposition 3.1 offers a way to sample the posterior distribution without retraining, by using the drifts (18) and (19) in the ODE (7) or the SDE (8).

### 3.4 Inference adaption

Suppose that we have learned the drifts $\eta_{0,1}$ in Definition 2.2 and wish to transport samples along a path $(\alpha_t, \beta_t)$ with fixed end points. We can leverage the flexibility of our formulation to perform *inference adaptation*, that is, optimize the path $(\alpha_t, \beta_t)$ used during generation to achieve specific objectives, such as minimizing computational cost, maximizing sample quality, or satisfying user constraints. This can be done in two ways: (1) *offline optimization*, where we pre-compute optimal paths for different scenarios using objectives like Wasserstein length minimization, and (2) *online adaptation*, where paths are dynamically adjusted during generation based on intermediate results or user feedback.

In the case of offline optimization, we could for example optimize the Wasserstein length of the path. That is, if we want to bridge the distributions $\mu_{\alpha_0,\beta_0}$ of $I(\alpha_0, \beta_0)$ and $\mu_{\alpha_1,\beta_1}$ of $I(\alpha_1, \beta_1)$ via $\mu_{\alpha_t,\beta_t}$ with $(\alpha_t, \beta_t) \in S$ for all $t \in [0,1]$ then the path that minimizes the Wasserstein length of the bridge distribution $\mu_{\alpha_t,\beta_t}$ solves

$$\min_{(\alpha_t,\beta_t)_t} \int_0^1 \mathbb{E}\big[\|\dot{\alpha}_t \eta_0(\alpha_t, \beta_t, I(\alpha_t, \beta_t)) + \dot{\beta}_t \eta_1(\alpha_t, \beta_t, I(\alpha_t, \beta_t))\|^2\big] dt \tag{22}$$

where the minimization is performed over paths $(\alpha_t, \beta_t)_t \equiv (\alpha_t, \beta_t)_{t \in [0,1]}$ such that $(\alpha_t, \beta_t) \in S$ for all $t \in [0,1]$ with their end points $(\alpha_0, \beta_0)$ and $(\alpha_1, \beta_1)$ prescribed and fixed.

---

**Algorithm 1:** Multitask learner

**input:** Samples $(x_0, x_1) \sim \mu$; choice of distribution $\nu(d\alpha, d\beta)$ and associated sampler.
**repeat**
> Draw batch $(x_0^i, x_1^i, \alpha_i, \beta_i)_{i=1}^M \sim \mu \times \nu$.
> Compute $I_i = \alpha_i x_0^i + \beta_i x_1^i$.
> Compute $\hat{L} = \frac{1}{M} \sum_{i=1}^M \|\hat{\eta}_0(\alpha_i, \beta_i, I_i) - x_0^i\|^2 + \|\hat{\eta}_1(\alpha_i, \beta_i, I_i) - x_1^i\|^2$.
> Take a gradient step on $\hat{L}$ to update $\hat{\eta}_0$ and $\hat{\eta}_1$.

**until** *converged*;
**output:** Drifts $\hat{\eta}_0$ and $\hat{\eta}_1$.

---

**Algorithm 2:** Multitask generator

**input:** Drifts $\hat{\eta}_0$, $\hat{\eta}_1$; choice of path $(\alpha_t \beta_t)_{t \in [0,1]}$ tailored to the generation task; data
> $I(\alpha_0, \beta_0) = \alpha_0 x_0 + \beta_0 x_1$; diffusion coefficient $\epsilon_t \geqslant 0$; time step $h = 1/K$ with $K \in \mathbb{N}$.

**initialize:** $\hat{X}_0^\epsilon = I(\alpha_0, \beta_0)$;
**for** $k = 0, \ldots, K-1$ **do**
> set $\hat{\eta}_0^k = \hat{\eta}_0(\alpha_{kh}, \beta_{kh}, \hat{X}_{kh}^\epsilon)$, $\hat{\eta}_1^k = \hat{\eta}_1(\alpha_{kk}, \beta_{kh}, \hat{X}_{kh}^\epsilon)$, and $z_k \sim N(0, \text{Id})$
> update $\hat{X}_{k+1k}^\epsilon = \hat{X}_{kh}^\epsilon + h\big(\dot{\alpha}_{kk} - \epsilon_{kh}\alpha_{kk}^{-1}\big)\hat{\eta}_0^k + h\dot{\beta}_{kh}\hat{\eta}_1^k + \sqrt{2\epsilon_{kh}h}\, z_k$,

**end**
**output:** $\hat{X}_1^\epsilon \stackrel{d}{=} I(\alpha_1, \beta_1)$ (approximately)

---

## 4 Algorithmic aspects

The algorithmic aspects of our framework can be summarized in a few key steps. First, we define a connected set $S$ of $(\alpha, \beta)$ such that the ensemble of different tasks we will want to perform correspond

to getting samples of $I(\alpha, \beta)$ at some value of $(\alpha_0, \beta_0) \in S$ and generating from them new data at another value of $(\alpha_1, \beta_1) \in S$. Second, we specify a measure $\nu$ on $S$ for the learning of the drifts $\eta_0$ and $\eta_1$ defined in (2). Third, we learn these drifts via minimization of the objectives in (3) and (4), using the procedure outlined in Algorithm 1. Note that we can possibly simplify this algorithm, learning only one of the two drifts and obtaining the other through the relation (5). Finally, given any pairs $(\alpha_0, \beta_0), (\alpha_1, \beta_1) \in S$, we use a path $(\alpha_t, \beta_t)_{t \in [0,1]}$ with $\alpha_t, \beta_t \in S$ for all $t \in [0, 1]$ and integrate the SDE (8) (or possibly the ODE (7) if we set $\epsilon_t = 0$) to perform the generation, as outlined in Algorithm 2. Note that this path could also be adapted on-the-fly during inference, using some feedback about the solution of the SDE.

# 5    Numerical experiments

Below we provide numerical realization of some of the various objectives that can be fulfilled with the multitask objective. Details of the experimental setup can be found in Appendix B.

## 5.1    Multitask inpainting and sequential generation

We evaluate our method on three datasets: MNIST, with images of size $28 \times 28$, CelebA, resized to $128 \times 128$, and of Animal FacesHQ focused on cat class, with images resized to $256 \times 256$. Details of the experimental setup are standard and can be found in Appendix B. In these experiments, we use the Hadamard interpolant (9).

**MNIST.** We demonstrate the versatility of our operator-based interpolant framework through inpainting and sequential generation tasks on MNIST. The results are shown Figure 3 where all the generated images come from the same model without any retraining.

For inpainting (left panels), we replace masked regions with Gaussian noise (shown as pink for clarity), then generate only these regions while preserving unmasked pixels. This is achieved by setting the entries of $\alpha$ to $1 - t$ for masked pixels and 0 for unmasked ones. To preserve unmasked pixels, we apply a secondary mask setting $\eta(\alpha, x)$ to zero at these positions.

Sequential generation (right panels) reformulates image creation as progressive inpainting. Starting with pure Gaussian noise, we generate the image block-by-block by successively updating the operator masks. Unlike single-pass inpainting, this requires multiple forward passes—one per block. For each pass, we apply $\alpha = 1 - t$ only to pixels in the current generation block, maintaining appropriate values for previously generated and remaining noise regions.

**CelebA and AFHQ-Cat.** We present benchmark results for all methods across various image restoration tasks, evaluating the average peak signal-to-noise ratio (PSNR) and structural similarity index (SSIM) on 100 test images from each dataset: AFHQ-Cat ($256 \times 256$) and CelebA ($128 \times 128$). To assess the performance of our methodology, we employed two types of masking: square masks of sizes $40 \times 40$ and $80 \times 80$ with added Gaussian noise of standard deviation 0.05, and random masks covering 70% of image pixels with Gaussian noise of standard deviation 0.01. We benchmark our method against four state-of-the-art interpolant-based restoration methods: PnP-Flow Martin et al. (2025), Flow-Priors Zhang et al. (2024), D-Flow Ben-Hamu et al. (2024), OT-ODE Pokle et al. (2024).

As shown in Table 1, our method consistently ranks either first or second in both reconstruction metrics across all tasks and datasets (with all values except the last row taken from Martin et al. (2025)). Regarding visual quality (Fig. 4), our method generates realistic, artifact-free images, albeit with slight over-smoothing at times.

## 5.2    Posterior sampling in the $\phi^4$-model

We apply our approach in the context of the $\phi^4$ model in $d = 2$ spacetime dimensions, a statistical lattice field theory where field configurations $\phi \in \mathbb{R}^{L \times L}$ represent the lattice state ($L$ denotes spatiotemporal extent)—for details see Appendix B.2. This model poses sampling challenges due to its phase transition from disorder to full order, during which neighboring sites develop strong correlations in sign and magnitude Vierhaus (2010); Albergo et al. (2019).

**Table 1:** PSNR and SSIM metrics for image inpainting methods on CelebA and AFHQ-Cat datasets.

| Method | CelebA | | | | AFHQ-Cat | | | |
| --- | --- | --- | --- | --- | --- | --- | --- | --- |
| | Random | | Block | | Random | | Block | |
| | PSNR | SSIM | PSNR | SSIM | PSNR | SSIM | PSNR | SSIM |
| Degraded | 11.82 | 0.197 | 22.12 | 0.742 | 13.35 | 0.234 | 21.50 | 0.744 |
| Pokle et al. (2024) | 28.36 | 0.865 | 28.84 | 0.914 | 28.84 | 0.838 | 23.88 | 0.874 |
| Ben-Hamu et al. (2024) | 33.07 | 0.938 | 29.70 | 0.893 | 31.37 | 0.888 | 26.69 | 0.833 |
| Zhang et al. (2024) | 32.33 | 0.945 | 29.40 | 0.858 | 31.76 | 0.909 | 25.85 | 0.822 |
| Martin et al. (2025) | 33.54 | 0.953 | **30.59** | **0.943** | 32.98 | 0.930 | 26.87 | 0.904 |
| Ours | **33.76** | **0.967** | 29.98 | 0.938 | **33.11** | **0.945** | **26.96** | **0.914** |

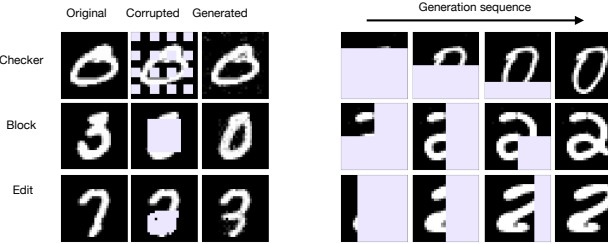

**Figure 3: Left**: In-painting on MNIST using various corruptions. **Right**: Image generation in arbitrary orders, starting from the same initial noise, with examples showing autoregressive, block-wise, and column-wise.

The $\phi^4$ model is specified by the following probability distribution

$$\mu(d\phi) = Z^{-1}e^{-E(\phi)}d\phi \tag{23}$$

where $Z = \int_{\mathbb{R}^{L \times L}} e^{-E(\phi)}d\phi$ is a normalization constant and $E(\phi)$ is an energy function defined as

$$E(\phi) = \frac{1}{2}\chi \sum_{a \sim b} |\phi(a) - \phi(b)|^2 + \frac{1}{2}\kappa \sum_a |\phi(a)|^2 + \frac{1}{4}\gamma \sum_a |\phi(a)|^4, \tag{24}$$

where $a, b \in [0, \ldots, L-1]^2$ denote the discrete positions on a 2-dimensional lattice of size $L \times L$, $a \sim b$ denotes neighboring sites on the lattice, and we assume periodic boundary conditions; $\chi > 0$, $\kappa \in \mathbb{R}$ and $\gamma > 0$ are parameters. We perform MCMC simulations to generate configuration in a parameter range close to the phase transition. We use these data to learn a stochastic interpolant of the form (9) which allows us to perform unconditional generation of new field configurations as well arbitrary inpainting (conditional generation given partially observed configurations), as reported in Appendix B.2. It also allows us to test the formalism of Section 3.3 and perform sampling of the posterior distribution defined by adding a applied field $h \in \mathbb{R}$ to the energy, i.e. using

$$E_r(\phi) = E(\phi) - h\sum_a \phi(a). \tag{25}$$

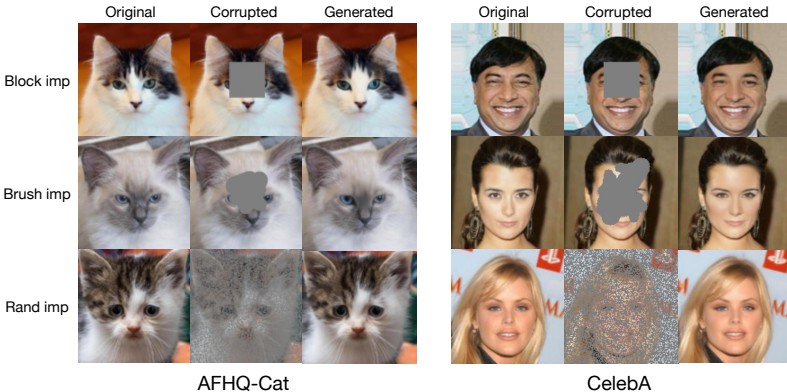

**Figure 4:** Inpainting using various masks **left panels**: AFHQ-Cat ($256 \times 256$). **Right panels**: CelebA ($128 \times 128$). Fixing block and random corruptions are scored against related works in Table 1, showing competitive or superior performance in all metrics.

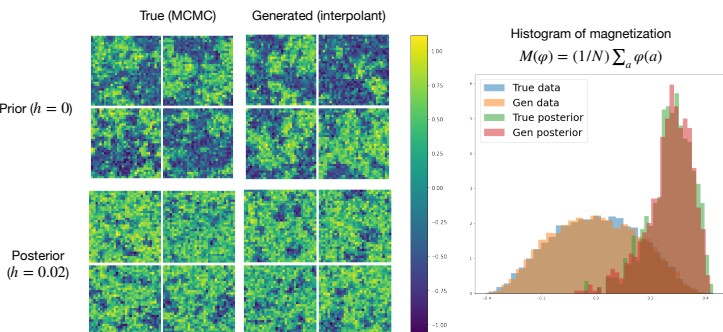

**Figure 5: Simulating a lattice $\phi^4$ theory. Top left**: $L = 32 \times L = 32$ lattice configurations at the phase transition. **Bottom left**: lattice examples with drift parameter $h = 0.02$. **Top middle**: Generated lattice examples at phase transition. **Bottom middle**: generated lattice examples with field $h = 0.02$. **Right**: magnetization of 2000 lattice configurations.

The additional term plays the role of a reward. The results of the generation based on Proposition 3.1 are shown on Figure 5, which indicate that our approach permits to valid sample configurations (as verified by their magnetization) of the posterior without retraining.

## 5.3 Planning and decision making in a maze

This section applies our framework to shortest path planning in maze environments, drawing from Janner et al. (2022) and Chen et al. (2024). Using the Hadamard product interpolant (9), we can impose that the paths pass through arbitrary locations in the maze (by setting $\alpha_i = 0$ at these locations), reformulating planning as a zero-shot inpainting problem. Unlike traditional reinforcement learning approaches that generate paths sequentially through Markov decision processes, our method therefore produces entire trajectories simultaneously. It also avoid additional guiding terms like Monte Carlo guidance used in Diffusion Forcing Chen et al. (2024).

For training, we use paths of length 300 randomly extracted from the trajectory of length 2,000,000 from Chen et al. (2024). For simplicity, we subsample these paths every six points, creating sparse paths of length 50, from which we can recover paths of length 300 through linear interpolation between consecutive points. At inference, we perform zero-shot generation between any two points in the maze by enforcing that the trajectory passes through these points: the length of the path between these locations can be varied by pinning the first point by setting $\alpha_1 = 0$, and the second point by setting $\alpha_i = 0$ with a value of $i \in [2, 50]$ that can be adjusted (see Appendix B.3 for details). Typical results are shown in Fig. 6. In terms of quality assessment, we check that the generated trajectories remain within allowed maze regions: all the 10,000 paths we generated between randomly chosen point pairs avoided the forbidden areas, demonstrating robust performance. More numerical experiments in Appendix B.3 demonstrate that with a similar strategy, one can impose the pathway to take detour at will, even if it implies generating a longer path.

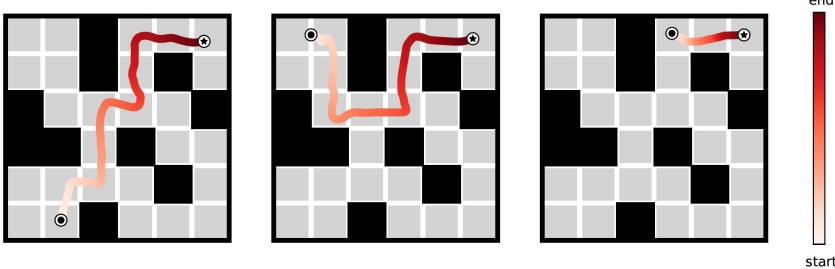

**Figure 6: One-shot generation of pathways between two arbitrary points in the maze.** The path length is automatically tuned via a simple heuristic, see discussion in Appendix B.3.

## Acknowledgements

We would like to thank Yilun Du for many helpful discussions on the maze planning problem. MSA is supported by a Junior Fellowship at the Harvard Society of Fellows as well as the National Science Foundation under Cooperative Agreement PHY-2019786 (The NSF AI Institute for Artificial Intelligence and Fundamental Interactions, http://iaifi.org).

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

## A Proofs

**Definition 2.2** (Multipurpose drift). *The* drifts $\eta_0, \eta_1 : S \times \mathcal{H} \to \mathcal{H}$ *are given by*

$$\eta_0(\alpha, \beta, x) = \mathbb{E}[x_0 | I(\alpha, \beta) = x], \qquad \eta_1(\alpha, \beta, x) = \mathbb{E}[x_1 | I(\alpha, \beta) = x], \tag{2}$$

*where* $\mathbb{E}[ \ \cdot \ | I(\alpha, \beta) = x]$ *denotes expectation over the coupling* $(x_0, x_1) \sim \mu$ *conditioned on the event* $I(\alpha, \beta) = x$.

**Lemma 2.3** (Drift objective). *Let* $\nu(d\alpha, d\beta)$ *be a probability distribution whose support is* $S$. *Then the drifts* $\eta_{0,1}(\alpha, \beta, x)$ *in Definition 2.2 can be characterized globally for all* $(\alpha, \beta) \in S$ *and all* $x \in \text{supp}(\mu_{\alpha,\beta})$ *via solution of the optimization problems*

$$\eta_0 = \underset{\hat{\eta}_0}{\text{argmin}} \, \mathbb{E}_{\substack{(\alpha,\beta)\sim\nu \\ (x_0,x_1)\sim\mu}} \left[ \|\hat{\eta}_0(\alpha, \beta, I(\alpha, \beta)) - x_0\|^2 \right], \tag{3}$$

$$\eta_1 = \underset{\hat{\eta}_1}{\text{argmin}} \, \mathbb{E}_{\substack{(\alpha,\beta)\sim\nu \\ (x_0,x_1)\sim\mu}} \left[ \|\hat{\eta}_1(\alpha, \beta, I(\alpha, \beta)) - x_1\|^2 \right], \tag{4}$$

*where* $\| \cdot \|$ *denotes the norm in* $\mathcal{H}$.

*Proof.* The lemma is a simple consequence of the $L^2$ characterization of the conditional expectation as least-squares-best predictor, see e.g. Section 9.3 in Williams (1991). $\qquad\square$

**Lemma 2.4** (Score). *Assume that* $\mathcal{H} = \mathbb{R}^d$ *and that the probability distribution* $\mu_{\alpha,\beta}$ *of the stochastic interpolant* $I(\alpha, \beta)$ *is absolutely continuous with respect to the Lebesgue measure with density* $\rho_{\alpha,\beta}(x)$. *Assume also that* $x_0 \sim N(0, Id)$ *and* $x_0 \perp x_1$. *Then the score* $s_{\alpha,\beta}(x) = \nabla \log \rho_{\alpha,\beta}(x)$ *is related to the drift* $\eta_0(\alpha, \beta, x)$ *via*

$$\eta_0(\alpha, \beta, x) = -\alpha s_{\alpha,\beta}(x). \tag{6}$$

*Proof.* The lemma follows from Stein's lemma (aka Gaussian integration by parts formula) that asserts that

$$\mathbb{E}[x_0 | I(\alpha, \beta, x) = x] = -\alpha s_{\alpha,\beta}(x) \tag{26}$$

as well as the definition of $\eta_0(\alpha, \beta, x)$ in (16). $\qquad\square$

**Proposition 2.5** (Probability flow). *Let* $(\alpha_t, \beta_t)_{t \in [0,1]}$ *be any one-parameter family of operators* $(\alpha_t, \beta_t) \in S$. *Assume that* $\alpha_t, \beta_t$ *are differentiable for all* $t \in [0, 1]$. *Then, for all* $t \in [0, 1]$, *the law of* $I(\alpha_t, \beta_t)$ *is the same as the law of the solution* $X_t$ *to*

$$\dot{X}_t = \dot{\alpha}_t \eta_0(\alpha_t, \beta_t, X_t) + \dot{\beta}_t \eta_1(\alpha_t, \beta_t, X_t), \qquad X_0 \overset{d}{=} I(\alpha_0, \beta_0). \tag{7}$$

*Proof.* From the framework of standard stochastic interpolants Albergo and Vanden-Eijnden (2022); Albergo et al. (2023a), we know that the law of $I_t = I(\alpha_t, \beta_t)$ is the same for all $t \in [0, 1]$ as the law of $X_t$, i.e. the solution to the probability flow ODE

$$\dot{X}_t = b_t(X_t), \qquad X_{t=0} \overset{d}{=} I_{t=0}, \tag{27}$$

where

$$b_t(x) = \mathbb{E}[\dot{I}_t | I_t = x].$$ (28)

By the chain rule $\dot{I}_t = \dot{\alpha}_t x_0 + \dot{\beta}_t x_1$ so that

$$b_t(x) = \dot{\alpha}_t \mathbb{E}[x_0 | I_t = x] + \dot{\alpha}_t \mathbb{E}[x_1 | I_t = x] \equiv \dot{\alpha}_t \eta_0(\alpha_t, \beta_t, x) + \dot{\beta}_t \eta_1(\alpha_t, \beta_t, x).$$ (29)

where $\eta_{0,1}$ are the drifts defined in (16). This means that (27) is (7). $\square$

**Proposition 2.6** (Diffusion). *Assume that $\mathcal{H} = \mathbb{R}^d$ and the probability distribution $\mu_{\alpha,\beta}$ of the stochastic interpolant $I(\alpha, \beta)$ is absolutely continuous with respect to the Lebesgue measure. Assume also that, in $I(\alpha, \beta)$, $x_0$ is Gaussian and $x_0 \perp x_1$. Then, under the same conditions as in Proposition 2.5, if $\alpha_t$ is invertible, for all $t \in [0,1]$ and any $\epsilon_t \geqslant 0$, the law of $I(\alpha_t, \beta_t)$ is the same as the law of the solution $X_t^\epsilon$ to*

$$dX_t^\epsilon = \left(\dot{\alpha}_t - \epsilon_t \alpha_t^{-1}\right)\eta_0(\alpha_t, \beta_t, X_t^\epsilon)dt + \dot{\beta}_t \eta_1(\alpha_t, \beta_t, X_t^\epsilon)dt + \sqrt{2\epsilon_t}dW_t, \quad X_0^\epsilon \overset{d}{=} I(\alpha_0, \beta_0).$$ (8)

*where $W_t$ is a Wiener process in $\mathbb{R}^d$.*

*Proof.* From the framework of standard stochastic interpolants Albergo and Vanden-Eijnden (2022); Albergo et al. (2023a), we know that the law of the solution $X_t$ to the probability flow ODE (27) is the same for all $t \in [0,1]$ as the law of $X_t^\epsilon$ solution to to SDE

$$dX_t^\epsilon = b_t(X_t)dt + \epsilon_t s_t(X_t^\epsilon)dt + \sqrt{2\epsilon_t}dW_t, \quad X_{t=0}^\epsilon, \overset{d}{=} I_{t=0},$$ (30)

where $s_t(x)$ is the score of the probability density function of $I_t = I(\alpha_t, \beta_t)$. Since $s_t(x) = s_{\alpha_t, \beta_t}(x)$, from Lemma 2.4, we have

$$s_t(x) = -\alpha_t^{-1}\eta_0(\alpha_t, \beta_t, x).$$ (31)

If we insert this expression in (30) and use (29), we see that this SDE reduces to (8). $\square$

**Proposition 3.1.** *Let*

$$\eta_0(\alpha, \beta, x) = \mathbb{E}[x_0 | I(\alpha, \beta) = x], \qquad \eta_1(\alpha, \beta, x) = \mathbb{E}[x_1 | I(\alpha, \beta) = x],$$ (16)

*be the drifts associated with the interpolant (14) and*

$$\eta_0^r(\alpha, \beta, x) = \mathbb{E}[x_0 | I_r(\alpha, \beta) = x], \qquad \eta_1^r(\alpha, \beta, x) = \mathbb{E}[x_1^r | I_r(\alpha, \beta) = x],$$ (17)

*be the drifts associated with the interpolant (15). If $\alpha$ and $\beta$ are invertible, then*

$$\eta_0^r(\alpha, \beta, x) = \alpha^{-1}\beta\beta_r^{-1}\alpha_r \eta_0(\alpha_r, \beta_r, x_r) + \alpha^{-1}(x - \beta\beta_r^{-1}x_r)$$ (18)
$$\eta_1^r(\alpha, \beta, x) = \eta_1(\alpha_r, \beta_r, x_r)$$ (19)

*as long as we can find a pair $(\alpha_r, \beta_r)$ that satisfies*

$$\beta_r^T \alpha_r^{-T} \alpha_r^{-1} \beta_r = \beta^T \alpha^{-T} \alpha^{-1} \beta - A$$ (20)

*and $x_r$ is given by*

$$x_r = \alpha_r \alpha_r^T \beta_r^{-T} \left(\beta^T \alpha^{-T} \alpha^{-1} x + b\right).$$ (21)

We will prove this proposition as a corollary of:

**Proposition A.1** (Posterior distributions). *Let $\mu_{\alpha,\beta}$ and $\mu_{\alpha,\beta}^r$ be the probability distributions of the stochastic interpolants defined in (14) and (15), respectively. Assume that $\alpha$ and $\beta$ are invertible, that the equations (20) for $\alpha_r$, $\beta_r$ in Proposition 3.1 have a solution, and that $x_r$ is given by (21). Then these distributions are related, up to a constant independent of $x$ and $x_r$, as*

$$\mu_{\alpha,\beta}^r(dx) = |\alpha_r||\alpha|^{-1}e^{R(\alpha,\beta,x)}\mu_{\alpha_r,\beta_r}(dx_r),$$ (32)

*where*

$$R(\alpha, \beta, x) = \tfrac{1}{2}|\alpha_r^{-1}x_r|^2 - \tfrac{1}{2}|\alpha^{-1}x|^2.$$ (33)

*Proof.* By definition of the probability distribution $\mu_{\alpha,\beta}^r(dx)$ of $I_r(\alpha,\beta)$, given any integrable and bounded test function $\phi : \mathbb{R}^d \to \mathbb{R}$ we have

$$\int_{\mathbb{R}^d} \phi(x)\mu_{\alpha,\beta}^r(dx) = \mathbb{E}[\phi(I_r(\alpha,\beta))]$$
$$= \int_{\mathbb{R}^d \times \mathbb{R}^d} \phi(\alpha x_0 + \beta x_1)(2\pi)^{-d/2} e^{-\frac{1}{2}|x_0|^2} dx_0 e^{r(x_1)}\mu_1(dx_1) \tag{34}$$

If instead of $x_0$ we use as new integration variable $x = \alpha x_0 + \beta x_1$, this becomes

$$\int_{\mathbb{R}^d} \phi(x)\mu_{\alpha,\beta}^r(dx) = |\alpha|^{-1} \int_{\mathbb{R}^d \times \mathbb{R}^d} \phi(x)(2\pi)^{-d/2} e^{-\frac{1}{2}|\alpha^{-1}(x-\beta x_1)|^2} dx e^{r(x_1)}\mu_1(dx_1). \tag{35}$$

Similarly, for the probability distribution $\mu_{\alpha,\beta}(dx)$ of $I(\alpha,\beta)$, we have

$$\int_{\mathbb{R}^d} \phi(x)\mu_{\alpha,\beta}(dx) = |\alpha|^{-1} \int_{\mathbb{R}^d \times \mathbb{R}^d} \phi(x)(2\pi)^{-d/2} e^{-\frac{1}{2}|\alpha^{-1}(x-\beta x_1)|^2} dx\mu_1(dx_1) \tag{36}$$

If in this equation we replace $\alpha$ by $\alpha_r$, $\beta$ by $\beta_r$, $x$ by $x_r$, and $\phi(x)$ by $\phi(x)e^{R(\alpha,\beta,x)}$, and multiply both side by $|\alpha_r|/|\alpha|$ it becomes:

$$|\alpha_r||\alpha|^{-1} \int_{\mathbb{R}^d} \phi(x)e^{R(\alpha,\beta,x)}\mu_{\alpha_r,\beta_r}(dx_r)$$
$$= |\alpha|^{-1} \int_{\mathbb{R}^d \times \mathbb{R}^d} \phi(x)e^{R(\alpha,\beta,x)}(2\pi)^{-d/2} e^{-\frac{1}{2}|\alpha_r^{-1}(x-\beta_r x_1)|^2} dx_r\mu_1(dx_1). \tag{37}$$

We can now require that the right hand side of (37) be the same as at the right hand-side of (35) (so that, $\mu_{\alpha,\beta}^r(dx) = |\alpha_r||\alpha|^{-1}e^{R(\alpha,\beta,x)}\mu_{\alpha_r,\beta_r}(dx_r)$), we arrive at the requirement that

$$-\frac{1}{2}|\alpha_r^{-1}(x_r - \beta_r x_1)|^2 + R(\alpha,\beta,x) = -\frac{1}{2}|\alpha^{-1}(x - \beta x_1)|^2 + \frac{1}{2}\langle x_1, Ax_1 \rangle + \langle b, x_1 \rangle, \tag{38}$$

where we used $r(x) = \frac{1}{2}\langle x, Ax \rangle + \langle b, x \rangle$. Since (38) must hold for all $x_1$, we can expand both sides of this equation, and equate the coefficient of order 2, 1 and 0 in $x_1$. They are completely equivalent to (20), (21), and (33), respectively. So as long as we can find solutions to (20), (32) holds. $\square$

*Proof of Proposition 3.1.* By definition of the conditional expectation, we have

$$\eta_1(\alpha,\beta,x) = \frac{\int_{\mathbb{R}^d} x_1 e^{-\frac{1}{2}|\alpha^{-1}(x-\beta x_1)|^2}\mu_1(dx_1)}{\int_{\mathbb{R}^d} e^{-\frac{1}{2}|\alpha^{-1}(x-\beta x_1)|^2}\mu_1(dx_1)}, \tag{39}$$

$$\eta_1^r(\alpha,\beta,x) = \frac{\int_{\mathbb{R}^d} x_1 e^{-\frac{1}{2}|\alpha^{-1}(x-\beta x_1)|^2} e^{r(x_1)}\mu_1(dx_1)}{\int_{\mathbb{R}^d} e^{-\frac{1}{2}|\alpha^{-1}(x-\beta x_1)|^2} e^{r(x_1)}\mu_1(dx_1)}. \tag{40}$$

If in the first equality we replace $\alpha$ by $\alpha_r$, $\beta$ by $\beta_r$, and $x$ by $x_r$, and assume that $\alpha_r$, $\beta_r$, and $x_r$ satisfy (38), by construction we obtain that (19) holds. To get (18), use (19) as well as relation (5) twice to deduce

$$\eta_0^r(\alpha,\beta,x) = \alpha^{-1}\big(x - \beta\eta_1^r(x,\alpha,\beta)\big)$$
$$= \alpha^{-1}\big(x - \beta\eta_1(x_r,\alpha_r,\beta_r)\big)$$
$$= \alpha^{-1}\big(x - \beta\beta_r^{-1}(x_r - \alpha_r\eta_0(x_r,\alpha_r,\beta_r))\big) \tag{41}$$
$$= \alpha^{-1}\beta\beta_r^{-1}\alpha_r\eta_0(x_r,\alpha_r,\beta_r) + \alpha^{-1}\big(x - \beta\beta_r^{-1}x_r\big).$$

$\square$

# B  Experimental details

Details for the experiments in Section 5 are provided here.

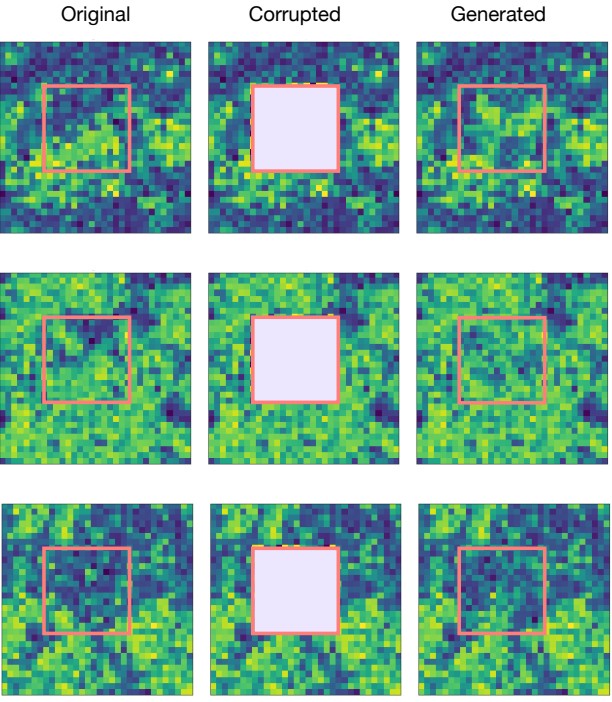

**Figure 7:** $\phi^4$**-model:** Inpainting of three different configurations.

## B.1 Multitask inpainting and sequential generation

For all image generation experiments, the U-Net architecture originally proposed in Ho et al. (2020) is used. The specification of architecture hyperparameters as well as training hyperparameters are given in Table 2. Training was done for 200 epochs on batches comprised of 30 draws from the target, and 50 time slices. The objectives given in 3 and 4 were optimized using the Adam optimizer. The learning rate was set to .0001 and was dropped by a factor of 2 every 1500 iterations of training. To integrate the ODE/SDE when drawing samples, we used a simple Euler integrator.

In order to progressively explore the space of the hypercube of $\alpha$ and $\beta$, we first learn a model in the diagonal of the hypercube, i.e where all entries of $\alpha$ are all the same value. We then fine-tune the first model for matrices $\alpha_t$ uniformly distributed in $[0,1]^d$. We also fine-tune the first model for matrices $\alpha_t$ decomposed by blocks of $4 \times 4$ where entries of each blocks contains the same values in $[0,1]^d$.

|  | MNIST | $\phi^4$ | CelebA | AFHQ-Cat |
|---|---|---|---|---|
| Dimension | 28×28 | 32×32 | 128×128 | 256×256 |
| # Training point | 60,000 | 100,000 | 190,000 | 5,000 |
| Batch Size | 50 | 100 | 128 | 64 |
| Training Steps | $4 \times 10^5$ | $2 \times 10^5$ | $4 \times 10^5$ | $4 \times 10^5$ |
| Attention Resolution | 64 | 64 | 64 | 64 |
| Learning Rate (LR) | 0.0002 | 0.0002 | 0.0001 | 0.0001 |
| LR decay (1k epochs) | 0.995 | 0.995 | 0.995 | 0.995 |
| U-Net dim mult | [1,2,2,2] | [1,2,2,2] | [1,2,4,8] | [1,2,4,8] |
| Learned $t$ embedding | Yes | Yes | Yes | Yes |
| # GPUs | 1 | 1 | 4 | 4 |

**Table 2:** Hyperparameters and architecture for MNIST, $\phi^4$ and maze datasets.

## B.2    Details about the $\phi^4$ Model

We define the discrete Fourier transform as

$$\hat{\phi}(k) = L^{-d/2} \sum_a e^{2i\pi k \cdot a/L} \phi(a) \quad \Leftrightarrow \quad \phi(a) = L^{-d/2} \sum_k e^{-2i\pi k \cdot a/L} \hat{\phi}(k), \qquad (42)$$

where $a, k \in [0, \dots, L-1]^d$, we can write the energy (24) as $E(\phi) = E_0(\phi) + U(\phi)$ with

$$E_0(\phi) = \hat{E}_0(\hat{\phi}) \equiv \frac{1}{2} \sum_k \hat{M}(k)|\hat{\phi}(k)|^2, \qquad \hat{M}(k) = 2\alpha \left( d - \sum_{\hat{e}} \cos(2\pi k \cdot \hat{e}/L) \right) + \beta_0, \quad (43)$$

where $\hat{e}$ denotes the $d$ basis vectors on the lattice and $\beta_0 > 0$ is an adjustable parameter; and

$$U(\phi) = \frac{1}{2}(\kappa - \kappa_0) \sum_a |\phi(a)|^2 + \frac{1}{4}\gamma \sum_a |\phi(a)|^4$$

$$\implies \hat{U}(\hat{\phi}) = \frac{1}{2}(\kappa - \kappa_0) \sum_k |\hat{\phi}(k)|^2 + \frac{1}{4}\gamma \sum_a \left| L^{-d/2} \sum_k e^{-2i\pi k \cdot a/L} \hat{\phi}(k) \right|^4. \qquad (44)$$

where $\phi$ and $\hat{\phi}$ are Fourier transform pairs as defined in (42): the last term can be implemented via $\sum_a (\text{ifft}(\hat{\phi}))^4(a)$.

**Sampling using the Langevin SDE:**    To obtain the ground-truth samples from the $\phi^4$ model, one option is to use the SDE

$$d\hat{\phi}_t(k) = -\hat{M}(k)\hat{\phi}_t(k)dt - (\kappa - \kappa_0)\hat{\phi}_t(k)dt - \gamma\widehat{\phi_t^3}(k)dt + \sqrt{2}d\hat{W}_t(k). \qquad (45)$$

where we denote

$$\widehat{\phi_t^3}(k) = L^{-d/2} \sum_a e^{2i\pi k \cdot a/L} \left( L^{-d/2} \sum_k e^{-2i\pi k \cdot a/L} \hat{\phi}_t(k) \right)^3 \qquad (46)$$

which can be implemented via $\text{fft}((\text{ifft}(\hat{\phi}_t))^3)$. This SDE may be quite stiff, however, a problem that can be alleviated by changing the mobility and using instead

$$d\hat{\phi}_t(k) = -\hat{\phi}_t(k)dt - (\kappa - \kappa_0)\hat{M}^{-1}(k)\hat{\phi}_t(k)dt - \gamma\hat{M}^{-1}(k)\widehat{\phi_t^3}(k)dt + \sqrt{2}\hat{M}^{-1/2}(k)d\hat{W}_t(k). \quad (47)$$

The discretized version of this equation reads

$$\hat{\phi}_{t_{n+1}}(k) = \hat{\phi}_{t_n}(k) - \Delta t_n \left( \hat{\phi}_{t_n}(k) + (\kappa - \kappa_0)\hat{M}^{-1}(k)\hat{\phi}_{t_n}(k) + \gamma\hat{M}^{-1}(k)\widehat{\phi_{t_n}^3}(k) \right)$$
$$+ \sqrt{2\Delta t_n}\hat{M}^{-1/2}(k)\hat{\eta}_n(k), \qquad (48)$$

where $\hat{\eta}_n$ is the Fourier transform of $\eta_n \sim N(0, \text{Id})$.

**Computing the generator of the posterior distribution**    As illustrated with (25), one would like to sample a slightly different $\phi^4$ model with energy function, noted $E_r$. $E$ and $E_r$ define respectively the prior and posterior distribution, via the Boltzmann's law (23). In this work, only relations of the form $E_r = E - \langle \phi, A\varphi \rangle - \langle h, \varphi \rangle$ are studied, combining a linear and quadratic term. $A$ is assumed to be definite negative.

**Lemma B.1.** *Assume that $\beta = 1 - \alpha$ and let*

$$\eta(\alpha, x) = \mathbb{E}[x_0 - x_1 | \alpha x_0 + (1 - \alpha)x_1 = x] = \eta_0(\alpha, 1 - \alpha, x) - \eta_1(\alpha, 1 - \alpha, x) \qquad (49)$$

*Then*

$$\eta_0(\alpha, 1 - \alpha, x) = x + (1 - \alpha)\eta(\alpha, x), \qquad (50)$$
$$\eta_1(\alpha, 1 - \alpha, x) = x - \alpha\eta(\alpha, x). \qquad (51)$$

*Proof of lemma B.1.*  Solving (5) and (49) in $(\eta_0, \eta_1)$ gives (50) and (51).    □

The idea is to use the drift of the prior to sample from the posterior. The following proposition makes it possible.

**Proposition B.2** (Posterior drift). *Assume $\beta = 1 - \alpha$ with $\alpha$ diagonal and invertible, and let*

$$\eta^r(\alpha, x) = \mathbb{E}\big[x_0 - x_1 | \alpha x_0 + (1 - \alpha)x_1^r = x\big] = \eta_0^r(\alpha, 1 - \alpha, x) - \eta_1^r(\alpha, 1 - \alpha, x) \tag{52}$$

*where the drifts $\eta_0^r$ and $\eta_1^r$ are defined in* (18) *and* (19), *respectively. Assume also that $A$ is diagonal, non-positive definite, and invertible. Then, $\beta_r = 1 - \alpha_r$ and $\eta^r$ can be expressed as:*

$$\eta^r(\alpha, x) = \alpha^{-1}\alpha_r \eta(\alpha_r, x_r) + \alpha^{-1}(x - x^r), \tag{53}$$

*where $\eta(\alpha, x)$ is the* drift *of the prior defined in* (49)*, and $\alpha_r$ and $x_r$ are given by*

$$\alpha_r = \alpha \frac{\sqrt{1 - 2\alpha + \alpha^2(1 - A)} - \alpha}{1 - 2\alpha - \alpha^2 A}, \tag{54}$$

$$x_r = \frac{\alpha_r^2(1 - \alpha)}{\alpha^2(1 - \alpha_r)} x + \frac{\alpha_r^2}{(1 - \alpha)} b, \tag{55}$$

*Proof of the Proposition B.2.* Using (50), (51) in (18) and (19), one obtains:

$$\eta_0^r(\alpha, \beta, x) = \alpha^{-1}\beta\beta_r^{-1}\alpha_r((1 - \alpha_r)\eta(\alpha_r, \beta_r, x_r) + x_r) + \alpha^{-1}(x - \beta\beta_r^{-1}x_r),$$
$$\eta_1^r(\alpha, \beta, x) = x_r - \alpha_r\eta(\alpha_r, \beta_r, x_r).$$

Then, take the difference, and regrouping terms together eventually yields:

$$\eta_0^r(\alpha, \beta, x) - \eta_1^r(\alpha, \beta, x) = \underbrace{(\alpha^{-1}\beta x_r + x_r)}_{=\alpha^{-1}\alpha_r}\eta(\alpha_r, \beta_r, x_r) + \underbrace{\alpha^{-1}\beta\beta_r^{-1}(\alpha_r - 1)x_r}_{=-\alpha^{-1}\beta x_r} - x_r + \alpha^{-1}x$$
$$= \alpha^{-1}\alpha_r\eta(\alpha_r, \beta_r, x_r) + \alpha^{-1}(x - x_r).$$

$\square$

**The linear case**    Assume for now that $A = 0$ and $b = h$. Note that one recovers the same case that in (25), that is, one applies a uniform magnetic field of magnitude $h$ over the whole lattice. What follows is simply a corollary of B.2.

**Proposition B.3.** *Assume $E_r = E - (\phi, h)$. Then, $\alpha_r = \alpha$, $\beta_r = \beta$, and*

$$\phi_r = \phi + \alpha\alpha^T\beta^{-T}h. \tag{56}$$

*Also, the posterior* drift *$\eta^r$ writes:*

$$\eta^r(\alpha, \beta, \phi) = \eta(\alpha, \beta, \phi_r) - \alpha^T\beta^{-T}h. \tag{57}$$

*Proof of Proposition B.3.* Since $A = 0$, (20) directly implies $\alpha_r = \alpha$ and $\beta_r = \beta$. Consequently, (56) follow from (21) and (57) from (53). $\square$

In summary, a simple shift proportional to $h$ appears in the posterior field $\phi_r$. It clearly tends to favor the alignment with the magnetic field, which obeys common sense.

**The quadratic case**

- Assume that $b = 0$ and $A = -k^2\text{Id}$.

In a similar fashion to the linear case, one derives analytical expressions for the quantities of interest.

**Proposition B.4.** *Assume $E_r = E - (\phi, A\phi) = E + k^2 \sum_a \phi(a)^2$, $\beta_r = 1 - \alpha_r$ and $\beta = 1 - \alpha$. Then*

$$\alpha_r = \alpha \frac{-\alpha + \sqrt{1 - 2\alpha + \alpha^2(1 + k^2)}}{1 - 2\alpha + \alpha^2 k^2}, \tag{58}$$

$$\phi_r = \frac{(-\alpha + \sqrt{1 - 2\alpha + \alpha^2(1 + k^2)})(1 - \alpha)}{\sqrt{1 - 2\alpha + \alpha^2(1 + k^2)}(1 - 2\alpha + \alpha^2 k^2)}\phi, \tag{59}$$

*and*

$$\eta^r(\alpha, \beta, \phi) = \frac{-\alpha + \sqrt{1 - 2\alpha + \alpha^2(1 + k^2)}}{1 - 2\alpha + \alpha^2 k^2} \eta(\alpha_r, \beta_r, \phi_r) \tag{60}$$

$$+ \alpha^{-1}\phi\left(1 - \frac{(-\alpha + \sqrt{1 - 2\alpha + \alpha^2(1 + k^2)})(1 - \alpha)}{\sqrt{1 - 2\alpha + \alpha^2(1 + k^2)}(1 - 2\alpha + \alpha^2 k^2)}\right). \tag{61}$$

*Proof of the Proposition B.4.* Given the assumptions, (20) yields:

$$(\text{Id} - \alpha_r)^T \alpha_r^{-T} \alpha_r^{-1} (\text{Id} - \alpha_r) = (\text{Id} - \alpha)^T \alpha^{-T} \alpha^{-1} (\text{Id} - \alpha) - k^2 \text{Id}.$$

Observing that $\alpha$ and $A$ are diagonals, $\alpha_r$ is also diagonal. Furthermore, assuming that $\alpha_r$ is proportional to the identity, the above reduces to the scalar equation (keeping the same notation for conciseness):

$$\frac{(1 - \alpha_r)^2}{\alpha_r^2} = \frac{(1 - \alpha)^2}{\alpha^2} + k^2$$

After a few elementary manipulations, one arrives at:

$$(1 - \alpha_r)^2 \alpha^2 = \alpha_r^2 (1 - \alpha)^2 + k^2 \alpha_r^2 \alpha^2.$$

This is a quadratic equation that admits two solutions. Only one is positive, and writes as:

$$\alpha_r = \frac{-\alpha^2 + \alpha\sqrt{1 - 2\alpha + \alpha^2(1 + k^2)}}{1 - 2\alpha + \alpha^2 k^2} = \alpha\frac{-\alpha + \sqrt{1 - 2\alpha + \alpha^2(1 + k^2)}}{1 - 2\alpha + \alpha^2 k^2}. \tag{62}$$

It is quite easy to check that the discriminant is always positive, so it does not pose any problem. Also, if $k \geqslant 0$ and $\alpha \in [0, 1]$, then $\alpha_r \in [0, 1]$. This property is necessary, since $\alpha \mapsto \eta(\cdot, \alpha)$ has been trained in the hypercube $[0, 1]^d$.

After elementary simplifications and recalling that $\beta_r = 1 - \alpha_r$ and (21), one has:

$$\phi_r = \frac{\alpha_r^2}{\alpha^2}\frac{1 - \alpha}{1 - \alpha_r} = \left(\frac{-\alpha + \sqrt{1 - 2\alpha + \alpha^2(1 + k^2)}}{1 - 2\alpha + \alpha^2 k^2}\right)^2 \frac{1 - \alpha}{1 - \alpha_r}.$$

Since $1 - \alpha_r = \frac{1 - 2\alpha + \alpha^2 k^2 + \alpha^2 - \alpha\sqrt{1 - 2\alpha + \alpha^2(1 + k^2)}}{1 - 2\alpha + \alpha^2 k^2} = \frac{1 - 2\alpha + \alpha^2(1 + k^2) - \alpha\sqrt{1 - 2\alpha + \alpha^2(1 + k^2)}}{1 - 2\alpha + \alpha^2 k^2}$, it yields:

$$\phi_r = \frac{(-\alpha + \sqrt{1 - 2\alpha + \alpha^2(1 + k^2)})^2}{1 - 2\alpha + \alpha^2 k^2}\frac{1 - \alpha}{1 - 2\alpha + \alpha^2(1 + k^2) - \alpha\sqrt{1 - 2\alpha + \alpha^2(1 + k^2)}}, \tag{63}$$

then factorizing by $\sqrt{1 - 2\alpha + \alpha^2(1 + k^2)}$ eventually gives (59).

Eventually, after replacing (62) and (59) into (53), (60) holds. $\square$

- Assume that $b = 0$ and $A = k^2 \text{Id}$.

In this case, the quadratic equation is:

$$\frac{(1 - \alpha_r)^2}{\alpha_r^2} = \frac{(1 - \alpha)^2}{\alpha^2} - k^2,$$

or otherwise stated:

$$(1 - \alpha_r)^2 \alpha^2 - (1 - \alpha)^2 \alpha_r^2 + \alpha_r^2 \alpha^2 k^2 = 0.$$

The discriminant of this polynomial is $\Delta = \alpha^2(1 - k^2) - 2\alpha + 1$. Assuming it strictly positive, among the two solutions, only one is positive:

$$\alpha_r = \alpha\frac{\sqrt{1 - 2\alpha + \alpha^2(1 - k^2)} - \alpha}{1 - 2\alpha - (\alpha k)^2}.$$

The polynomial inside the square root is positive if and only if $\alpha \notin [\frac{1}{1+k}, \frac{1}{1-k}]$. To see that, see there exists always two real roots, since the discriminants is $4k^2 > 0$. Those roots are $\frac{1}{1+k} < 1$ and $\frac{1}{1-k} > 1$. Since $\alpha_r \in [0, 1]$ must be respected for all $\alpha \in [0, 1]$, only $k < 1$ can be considered with our method. Consequently, sampling using stochastic interpolants from $\alpha = 1$ to $\alpha = 0$ appears impossible with this method.

## B.3 Details about the maze experiment

We use the Hadamard interpolant (9) and estimate the drift $\eta(\alpha, x)$ defined in (10) by approximating it with a U-Net neural network Ho et al. (2020), trained with an Adam optimizer Kingma and Ba (2017). The U-Net comprises 4 stages with $48, 80, 160$, and $256$ channels respectively for the encoding flow. The decoder has the same architecture as the encoder but in reverse order, with added residual connections Ronneberger et al. (2015). Each stage consists of 2 residual blocks, with the first concatenated with a self-attention block. The input vector has shape $d \times 2$, where row $i$ contains the $x$ and $y$ coordinates of the $i$-th point in the trajectory.

In contrast to conventional U-Net architectures, we perform interpolation and max pooling operations independently on each coordinate column to increase and reduce dimensions only along the trajectory length axis. The convolution kernel size is $5 \times 2$, processing each point's coordinates together with those of its two temporal predecessors and successors in the sequence. We add the necessary padding to maintain identical input and output dimensions, which amounts to padding by two rows at the top and bottom of the input vector.

Given a pair of randomly chosen points in the maze, we must determine where to constrain these points along the generated trajectory. If the constraint points are placed too far apart in the sequence (large index difference), the resulting path will likely not be the shortest route; conversely, if placed too close together (small index difference), the generated path has an increased chance of cutting through forbidden regions, making it inadmissible. To address this trade-off, we adopt the following heuristic. We fix the starting point at the beginning of the path (index $i = 1$) and employ a progressive search for the target point placement using the candidate indices $[5, 10, 20, 30, 40, 45, 50]$. We first generate a path with the target point constrained at index 5 (creating a short trajectory). If this path intersects forbidden regions, we increase the target index to 10 (allowing a longer path), and continue this process until we generate a valid path that successfully avoids all obstacles.

On Figure 8, we impose paths to go by the bottom-right corner, the constraint is visible as a small white dot. The path length adapts accordingly.

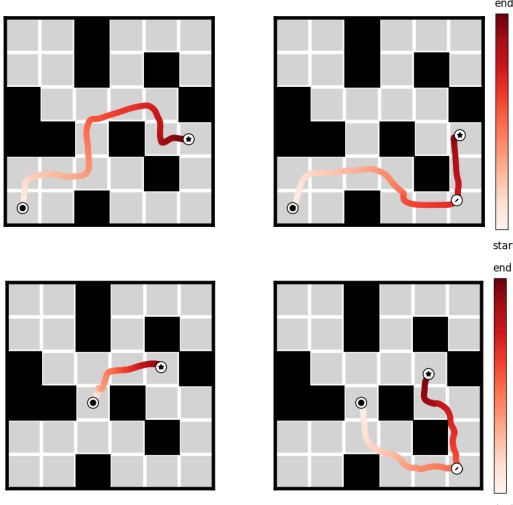

**Figure 8:** Two generated pathways, one per row. *Left*: No constraint is imposed on the path, other than joining the two endpoints. *Right*: An additional constraint is imposed: the path must pass through the bottom-right corner, represented by a white dot.

## C   Additional experimental results

Here we provide additional infilling image results, given in Figure 9.

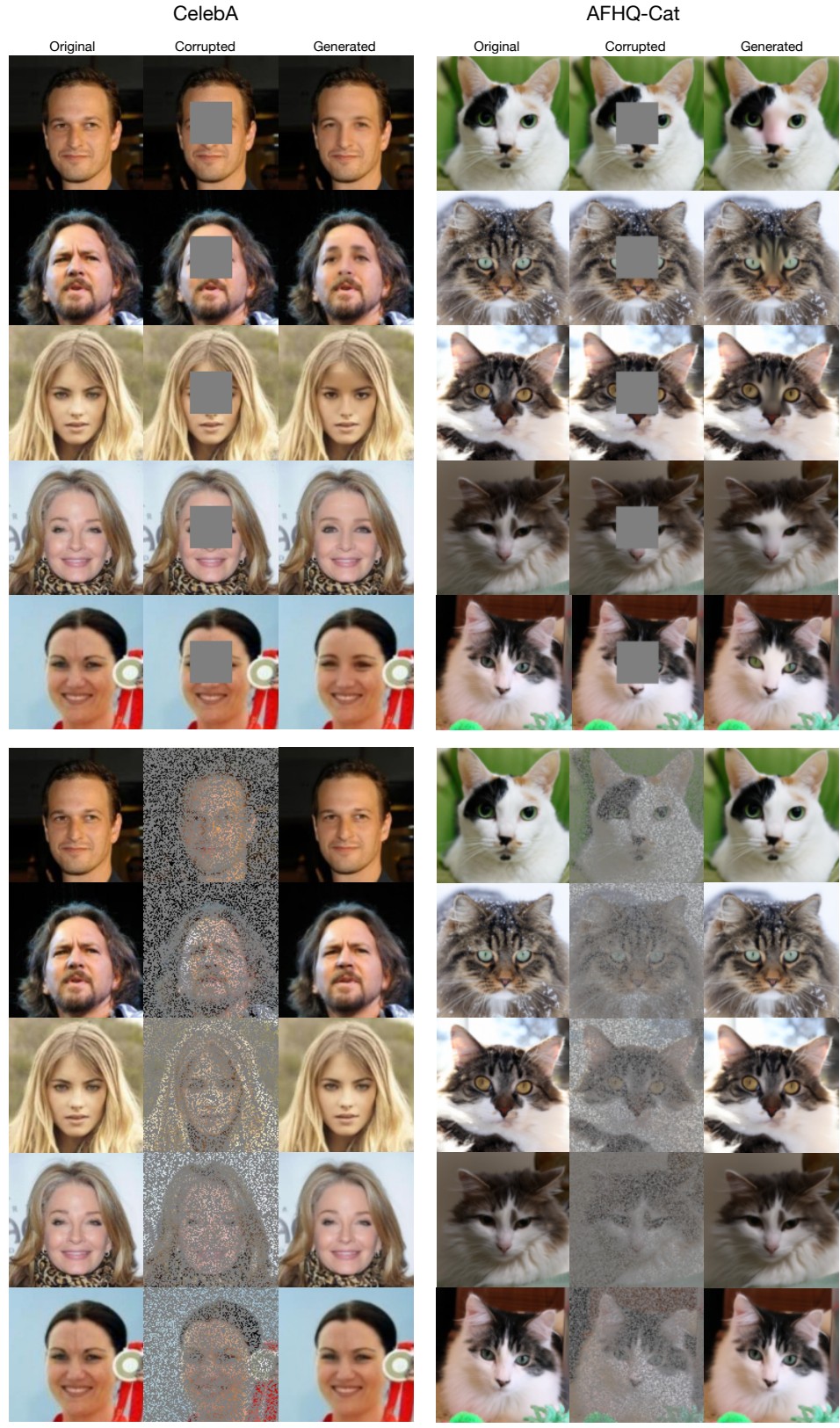

**Figure 9:** Additional images demonstrating the inpainting task: Block inpainting is shown in the top panels, while random inpainting is displayed in the bottom panels. The left panels depict images from the AFHQ dataset, and the right panels show images from the CelebA dataset.

# NeurIPS Paper Checklist

The checklist is designed to encourage best practices for responsible machine learning research, addressing issues of reproducibility, transparency, research ethics, and societal impact. Do not remove the checklist: **The papers not including the checklist will be desk rejected.** The checklist should follow the references and follow the (optional) supplemental material. The checklist does NOT count towards the page limit.

Please read the checklist guidelines carefully for information on how to answer these questions. For each question in the checklist:

- You should answer [Yes] , [No] , or [NA] .
- [NA] means either that the question is Not Applicable for that particular paper or the relevant information is Not Available.
- Please provide a short (1–2 sentence) justification right after your answer (even for NA).

**The checklist answers are an integral part of your paper submission.** They are visible to the reviewers, area chairs, senior area chairs, and ethics reviewers. You will be asked to also include it (after eventual revisions) with the final version of your paper, and its final version will be published with the paper.

The reviewers of your paper will be asked to use the checklist as one of the factors in their evaluation. While "[Yes] " is generally preferable to "[No] ", it is perfectly acceptable to answer "[No] " provided a proper justification is given (e.g., "error bars are not reported because it would be too computationally expensive" or "we were unable to find the license for the dataset we used"). In general, answering "[No] " or "[NA] " is not grounds for rejection. While the questions are phrased in a binary way, we acknowledge that the true answer is often more nuanced, so please just use your best judgment and write a justification to elaborate. All supporting evidence can appear either in the main paper or the supplemental material, provided in appendix. If you answer [Yes] to a question, in the justification please point to the section(s) where related material for the question can be found.

IMPORTANT, please:

- **Delete this instruction block, but keep the section heading "NeurIPS Paper Checklist",**
- **Keep the checklist subsection headings, questions/answers and guidelines below.**
- **Do not modify the questions and only use the provided macros for your answers**.

