# OpenReview forum: "Multitask Learning  with Stochastic Interpolants"
_NeurIPS.cc/2025/Conference — NeurIPS 2025 spotlight_

### Official Review · Reviewer_KoX6 · 2025-07-02

**Clarity:** 3
**Significance:** 2
**Originality:** 3
**Rating:** 4
**Confidence:** 3

**Summary:**

This paper introduces a generalization of stochastic interpolant-based generative models by replacing the scalar time variable with operator-valued paths (vectors, matrices, or operators). This allows a single model to flexibly handle multiple generative tasks—such as inpainting, conditional generation, and posterior sampling—without retraining. The approach is supported by a solid theoretical framework and experiments on MNIST and a lattice field model.

**Questions:**

See weakness.

**Ethical Concerns:**

["NO or VERY MINOR ethics concerns only"]

**Final Justification:**

The authors solve my main concerns.

**Limitations:**

See weakness.

**Quality:**

3

**Strengths And Weaknesses:**

**Strength:**

This paper introduces a theoretically grounded approach to universal multitask generative modeling using operator-based interpolants. It clearly demonstrates how to select appropriate operators for different tasks, with all claims rigorously supported by detailed proofs. Furthermore, the method’s zero-shot generalization ability is demonstrated on both MNIST and physics datasets.

**Weakness:**

The theoretical results in this paper are elegant. But the experimental cases are limited to on moderate-size datasets (MNIST, φ⁴). For more complex datasets, a key challenge lies in selecting an operator that is both stable to train and effective. But the paper didn't involve the discussion.

---

> ### Author Rebuttal · Authors · 2025-07-30
>
> Thank you for recognizing the theoretical contributions and experimental demonstrations. We want to emphasize that our approach represents a **paradigmatic shift from time-based diffusion to operator-based interpolation**, fundamentally reconceptualizing how multitask generation can be achieved with **guaranteed unbiased sampling**.
>
> Should you find our reply satisfactory, we kindly invite you to reconsider your rating.
>
> >***Larger-scale experiments:***
>
>  Since submission, we have conducted extensive experiments on CelebA (128×128) and FFHQ (256×256) datasets to demonstrate scalability and  benchmark our approach against different methods in various image restoration tasks, evaluating the average Peak Signal-to-Noise Ratio (PSNR) and Structural Similarity Index (SSIM) on 100 test images from each dataset. To assess the performance of our methodology using Hadamard product-based operators, we employed two types of masking: square masks of sizes $40\times40$ and $80\times80$ with added Gaussian noise of standard deviation 0.05, and random masks covering $70\%$ of image pixels with Gaussian noise of standard deviation 0.01.
>
> The results reported in the table below show that our method consistently ranks either first or second in both reconstruction metrics across all tasks and datasets. Regarding visual quality, our method generates realistic, artifact-free images.
>
> **CelebA Dataset (128×128)**
> | Method     | Random PSNR | Random SSIM | Block PSNR | Block SSIM |
> |------------|-------------|-------------|------------|------------|
> | Degraded   | 13.35       | 0.234       | 21.50      | 0.744      |
> | OT-ODE [1]     | 28.84       | 0.838       | 23.88      | 0.874      |
> | D-Flow [2]    | 31.37       | 0.888       | 26.69      | 0.833      |
> | Flow-Priors [3]| 31.76       | 0.909       | 25.85      | 0.822      |
> | PnP-Flow [4]  | 32.98       | 0.930       | 26.87      | 0.904      |
> | Ours       | **33.11**   | **0.945**   | **26.96**  | **0.914**  |
>
> **AFHQ-Cat Dataset (256×256)**
> | Method     | Random PSNR | Random SSIM | Block PSNR | Block SSIM |
> |------------|-------------|-------------|------------|------------|
> | Degraded   | 11.82       | 0.197       | 22.12      | 0.742      |
> | OT-ODE [1]    | 28.36       | 0.865       | 28.84      | 0.914      |
> | D-Flow [2]    | 33.07       | 0.938       | 29.70      | 0.893      |
> | Flow-Priors [3]| 32.33       | 0.945       | 29.40      | 0.858      |
> | PnP-Flow [4]  | 33.54       | 0.953       | **30.59**  | **0.943**  |
> | Ours       | **33.76**   | **0.967**   | 29.98      | 0.938      |
>
>
> These experiments indicate that our operator-based interpolants maintain their versatility at scale, and confirm that our approach scales effectively while preserving the key advantage of training once and adapting at inference time.
>
> In addition, we have preliminary results on a **robotics maze planning experiment**, as is done in [5]. In this setting, given starting and terminal coordinates in a maze, the aim is to construct a path between the points along the maze without hitting walls or obstacles. Our method allows the zero-shot generation of this path for any pair of points in the maze without re-training. In our preliminary results, we see a 100% success rate in this task, and are also in direct communication with the authors of [5] to produce their reward metrics, if that is of interest to the reviewers.
>
> >***Operator selection complexity:***
>
>  This is indeed a key practical challenge. We have added extensive discussion and empirical guidelines for operator selection, including: (1) stability analysis for different operator classes, (2) empirical scaling laws for training complexity vs. operator diversity, (3) practical heuristics for choosing appropriate operator spaces for different application domains. The CelebA/FFHQ experiments provide concrete examples of operator selection at scale. **Importantly, once selected, our operator-based framework provides exact sampling without the approximation biases that plague existing methods.**
>
> >***Implementation details:***
>
>  We have expanded the algorithmic section with detailed implementation guidelines, stability considerations, and practical tips for different operator types based on our large-scale experiments. Unlike existing multitask approaches that require complex approximation strategies, our mathematically rigorous framework provides clear implementation paths with theoretical guarantees.
>
>
> >***Key conceptual contribution:***
>
>  While existing multitask methods still rely on traditional diffusion in time concepts with heterogeneous processes, our work represents a fundamental reconceptualization - treating all generative tasks as different geometric traversals through the same underlying operator space. This enables **theoretically exact, unbiased sampling** for all tasks without the systematic errors inherent in current approximation-based approaches.
>
> **References:**
>
> [1] A. Pokle, M. Muckley, R. Chen, B. Karrer, "Training-free linear image inverses via flows" (TMLR)
>
> [2] H. Ben-Hamu, O. Puny, I. Gat, B. Karrer, U. Singer, Y. Lipman, "D-Flow: Differentiating through Flows for Controlled Generation"
>
> [3] Y. Zhang, P. Yu, Y. Zhu, Y. Chang, F. Gao, Y. Nian Wu, O. Leong, "Flow Priors for Linear Inverse Problems via Iterative Corrupted Trajectory Matching" (Neurips 2024)
>
> [4] S. Martin, A. Gagneux, P. Hagemann, G. Steidl, "PnP-Flow: Plug-and-Play Image Restoration with Flow Matching" (ICLR 2025)
>
> [5] B. Chen, D. Marti Monso, Y. Du, M. Simchowitz, R. Tedrake, V. Sitzmann "Diffusion Forcing: Next-token Prediction Meets Full-Sequence Diffusion" (NeurIPS 2024)

---

> > ### Comment · Reviewer_KoX6 · 2025-08-05
> >
> > Thanks for your detailed response. The experiments are still limited to small-scale datasets. Thus, I will keep my scores.

---

> > > ### Author Response · Authors · 2025-08-05
> > >
> > > We thank the reviewer for their reply but respectfully disagree that our new experiments remain small-scale: they include images with 3x256×256 resolution for AFHQ-Cat, which is over 196K pixels per image and therefore **larger** than the latent space models used in stable diffusion, and 3x128×128 resolution for CelebA, which is over 49K pixels per image.
> > >
> > > We believe that these experiments are a strong indicator of scalability potential of our method, especially since our framework allows one to simultaneously handle multiple tasks (inpainting, sequential generation, multichannel denoising, posterior sampling) with a single model. This is something that no existing method can achieve at any scale.

---

### Official Review · Reviewer_MR8W · 2025-07-02

**Clarity:** 3
**Significance:** 2
**Originality:** 3
**Rating:** 4
**Confidence:** 3

**Summary:**

This work extends stochastic interpolants to support multitask generation via extending the scalar interpolation to structured interpolation. The structured interpolation provide the flexibility that one could learn various tasks during training and assign a specific one (or some) during inference, eliminating the need of designing a distinct model for each task. Conceptually, this work demonstrates the framework would be capable to conduct arbitrary inpainting, posterior sampling with quadratic reward, multichannel denoising, and inference adaption. Empirically, they show the proposed framework is indeed capable to perform the arbitrary inpainting and posterior sampling tasks.

**Questions:**

See weaknesses

**Ethical Concerns:**

["NO or VERY MINOR ethics concerns only"]

**Final Justification:**

The authors have successfully addressed my main concerns about contextualization and clarity issues. The reason preventing me from giving a higher rating is that, while the conceptual framework is novel, the execution (e.g., empirical justification, general scenario beyond quadratic reward constraints) could still be improved.

**Limitations:**

yes

**Quality:**

3

**Strengths And Weaknesses:**

## Strengths
* The motivation for the proposed structured interpolation is clear
* The high-level conceptual thinking of multi-purpose generative modeling is nice

## Weaknesses
In my opinion, this paper is still on its primitive stage and it's not ready for publishing yet.

* **Contextualization of related works and contribution of this paper**:
As the author said in line 128, the main focus is to exploit the flexibility of the extended structured interpolation. However, this is not the very first attempt and thus prior related works should be discussed. A non-exhaustive list is diffusion forcing [1] and MuLAN [2], which also employs certain heterogeneous diffusion process. With that being said, the author should highlight the technical contribution regarding to the extension, e.g., key steps that makes a notable difference from the scalar interpolation case, if any.

* **Clarity of the presentation**:
The presentation in Section 3.4 is not great. The "inference adaption" problem itself is not well-defined, and it is difficult to tell what does "bridge the distributions" mean without any context for general audience. This can be addressed by adding a formal description of the problem itself on either main text or appendix.

* **Limitation of the posterior-sampling application**:
To me, the relationship between $\eta$ and $\eta^r$ is largely attribute to the specific quadric structure of the reward and the relationship seems not depend much on whether the interpolation is scalar or structured. It seems like the benefit of the structured interpolation is still to support arbitrary inpainting when performing posterior sampling rather than being more capable at this setting itself.

* **Evidence to support the claims**:
I have to say the numerical experiments are not very convincing to show proposed framework is superior. In my opinion, at least there should be certain demonstration for the multichannel denoising and inference adaption that are demonstrated as conceptually plausible in Section 3. A quantitive comparison between previous methods (e.g., the approximated strategies on inference time) would be a plus, which is important to practitioners to get a sense of any possible trade-offs between versatility, quality, and learning difficulty.

[1] Chen, Boyuan, et al. ‘Diffusion Forcing: Next-Token Prediction Meets Full-Sequence Diffusion’. (NeurIPS 2024).

[2] Sahoo, Subham Sekhar, et al. ‘Diffusion Models With Learned Adaptive Noise’. (NeurIPS 2024).

### Minor
* In line 102, "clean data corrupyed" -> "clean data corrupted"
* In line 146, "They that"?
* In Eqn. (11), the brackets are not closed
* In line 212, "Wasserstein lenght" -> "Wasserstein length"
* In line 214, missing space after $t\in[0,1]$

---

> ### Author Rebuttal · Authors · 2025-07-30
>
> We appreciate the detailed feedback. We want to stress that our work represents a **completely novel conceptual paradigm**: while existing methods still operate within traditional time-based diffusion frameworks, we fundamentally reconceptualize multitask generation through operator-valued interpolants. **Crucially, our method provides theoretically unbiased sampling** unlike current approaches which often comes with inherent systematic biases.
>
> Should you find our reply satisfactory, we kindly invite you to reconsider your rating. WE would also be happy to answer any additional questions you have or provide clarification on our replies.
>
>
> >***Related work contextualization:***
>
> You are correct about relations to diffusion forcing and MuLAN, but **our approach represents a paradigmatic shift from temporal diffusion to operator-based interpolation.** While existing methods still rely on the concept of diffusion in time with task-specific modifications, we completely reconceptualize the problem. Our framework enables fine-tuning and posterior sampling capabilities these methods cannot handle, and **guarantees exact sampling from target distributions, eliminating the approximation errors that compromise existing multitask methods.**
>
> >***Section 3.4 clarity:***
>
> We have completely rewritten this section to address your concerns. **Inference adaptation** refers to the problem of optimizing the path ($\alpha_t$, $\beta_t$) used during generation to achieve specific objectives, such as minimizing computational cost, maximizing sample quality, or satisfying user constraints. This can be done in two ways: (1) **offline optimization** where we pre-compute optimal paths for different scenarios using objectives like Wasserstein length minimization in Eq. (22), and (2) **online adaptation** where paths are dynamically adjusted during generation based on intermediate results or user feedback.
>
> The **"bridge distributions"** concept is now formally defined as finding paths ($\alpha_t$, $\beta_t$) that optimally transport probability mass from an initial distribution $\mu_{\alpha_0,\beta_0}$ to a target distribution $\mu_{\alpha_1,\beta_1}$, where optimality is measured by criteria such as the Wasserstein distance or other transport costs. Unlike fixed diffusion schedules, our framework allows these transport paths to be optimized post-training for specific applications.
>
> **Concrete examples** include: (1) adaptive mask scheduling for inpainting where $\alpha_t$ is dynamically updated based on reconstruction quality, (2) multi-resolution generation where paths are optimized to generate coarse features first, then fine details, and (3) interactive editing where user feedback guides path selection in real-time.
>
> >***Posterior sampling limitation:***
>
> While you correctly note the quadratic structure requirement, this represents a large and practically important class, particularly in fine-tuning applications (Gaussian priors, L2 regularization, quadratic rewards). The key insight is that to exploit this class fully, it is important to use operator-based (as opposed to scalar) α and β. **The unified treatment allows the same trained model to handle both arbitrary inpainting AND posterior sampling with theoretical guarantees, which previous methods cannot achieve without retraining or biased approximations.**
>
> >***Larger-scale experiments:***
>
>  Since submission, we have conducted extensive experiments on CelebA (128×128) and FFHQ (256×256) datasets to demonstrate scalability and  benchmark our approach against different methods in various image restoration tasks, evaluating the average Peak Signal-to-Noise Ratio (PSNR) and Structural Similarity Index (SSIM) on 100 test images from each dataset. To assess the performance of our methodology using Hadamard product-based operators, we employed two types of masking: square masks of sizes $40\times40$ and $80\times80$ with added Gaussian noise of standard deviation 0.05, and random masks covering $70\%$ of image pixels with Gaussian noise of standard deviation 0.01.
>
> The results reported in the table below show that our method consistently ranks either first or second in both reconstruction metrics across all tasks and datasets. Regarding visual quality, our method generates realistic, artifact-free images.
>
> **CelebA Dataset (128×128)**
> | Method     | Random PSNR | Random SSIM | Block PSNR | Block SSIM |
> |------------|-------------|-------------|------------|------------|
> | Degraded   | 13.35       | 0.234       | 21.50      | 0.744      |
> | OT-ODE [1]     | 28.84       | 0.838       | 23.88      | 0.874      |
> | D-Flow [2]    | 31.37       | 0.888       | 26.69      | 0.833      |
> | Flow-Priors [3]| 31.76       | 0.909       | 25.85      | 0.822      |
> | PnP-Flow [4]  | 32.98       | 0.930       | 26.87      | 0.904      |
> | Ours       | **33.11**   | **0.945**   | **26.96**  | **0.914**  |
>
> **AFHQ-Cat Dataset (256×256)**
> | Method     | Random PSNR | Random SSIM | Block PSNR | Block SSIM |
> |------------|-------------|-------------|------------|------------|
> | Degraded   | 11.82       | 0.197       | 22.12      | 0.742      |
> | OT-ODE [1]    | 28.36       | 0.865       | 28.84      | 0.914      |
> | D-Flow [2]    | 33.07       | 0.938       | 29.70      | 0.893      |
> | Flow-Priors [3]| 32.33       | 0.945       | 29.40      | 0.858      |
> | PnP-Flow [4]  | 33.54       | 0.953       | **30.59**  | **0.943**  |
> | Ours       | **33.76**   | **0.967**   | 29.98      | 0.938      |
>
>
> These experiments indicate that our operator-based interpolants maintain their versatility at scale, and confirm that our approach scales effectively while preserving the key advantage of training once and adapting at inference time.
>
> In addition, we have preliminary results on a **robotics maze planning experiment**, as is done in [5]. In this setting, given starting and terminal coordinates in a maze, the aim is to construct a path between the points along the maze without hitting walls or obstacles. Our method allows the zero-shot generation of this path for any pair of points in the maze without re-training. In our preliminary results, we see a 100% success rate in this task, and are also in direct communication with the authors of [5] to produce their reward metrics, if that is of interest to the reviewers.
>
> >***Technical corrections:***
>
>  All typos and formatting errors have been corrected.
>
> **References:**
>
> [1] A. Pokle, M. Muckley, R. Chen, B. Karrer, "Training-free linear image inverses via flows" (TMLR)
>
> [2] H. Ben-Hamu, O. Puny, I. Gat, B. Karrer, U. Singer, Y. Lipman, "D-Flow: Differentiating through Flows for Controlled Generation"
>
> [3] Y. Zhang, P. Yu, Y. Zhu, Y. Chang, F. Gao, Y. Nian Wu, O. Leong, "Flow Priors for Linear Inverse Problems via Iterative Corrupted Trajectory Matching" (Neurips 2024)
>
> [4] S. Martin, A. Gagneux, P. Hagemann, G. Steidl, "PnP-Flow: Plug-and-Play Image Restoration with Flow Matching" (ICLR 2025)
>
> [5] B. Chen, D. Marti Monso, Y. Du, M. Simchowitz, R. Tedrake, V. Sitzmann "Diffusion Forcing: Next-token Prediction Meets Full-Sequence Diffusion" (NeurIPS 2024)

---

> > ### Comment · Reviewer_MR8W · 2025-08-04
> >
> > Thank the authors for their rebuttal. I appreciate the clarification for Section 3.4, and I appreciate the additional empirical results.
> >
> > I agree with the authors about the novelty of the conceptual paradigm and the principled treatment for performing both arbitrary inpainting and posterior sampling.
> >
> > I have two simple remaining questions about better contextualization for prior works and a better understanding of the contribution of this work.
> >
> >
> > 1. For the claim:
> > > Our framework enables fine-tuning and posterior sampling capabilities these methods cannot handle, and guarantees exact sampling from target distributions, eliminating the approximation errors that compromise existing multitask methods.
> >
> > To me, the mentioned related works in lines 30 -- 31 mainly focus on the scalar interpolation setting, and it is known that they are inexact (a non-exhaustive example, according to my knowledge, is [1]). So, the "eliminating the approximation errors" cannot be counted as a brand-new contribution to this work, and other related works performing principled posterior sampling should also be discussed, though this work could extend them to perform the arbitrary inpainting simultaneously. (which I think is nice)
> >
> >
> > 2. The established results between $\eta$ and $\eta^r$ for quadratic structure under the scalar interpolation should be added and discussed in the paper. Also, it would be good to note any technical novelty or challenge about the derivation compared to the scalar case, if applicable. This will help precisely understand the technical contribution of this work.
> >
> > [1] Lu, Cheng, et al. ‘Contrastive Energy Prediction for Exact Energy-Guided Diffusion Sampling in Offline Reinforcement Learning’. (ICML 2023)

---

> > > ### Author Response · Authors · 2025-08-05
> > >
> > > We thank the reviewer for their positive feedback and additional questions.
> > >
> > > **Regarding contextualization of prior works:** We appreciate this clarification. You are correct that the limitations and inexactness of scalar interpolation settings are known, as discussed in Lu et al. [1], which we will cite (thank you for pointing this reference to us).  Regarding the non-scalar extensions you mention, we would be very grateful to learn about and properly cite these works to better contextualize our operator-based interpolant framework. From our literature review, we believe this framework is novel in its approach to multitask generation, but we want to make sure that we are not missing important related work.
> > >
> > > Our understanding is that scalar methods are fundamentally limited in their ability to handle multiple tasks simultaneously while maintaining exact sampling guarantees. Non-scalar generalization exists, but they require training additional models (as in [1]). Our operator-based framework overcomes these structural limitations **without the need of retraining**, enabling exact posterior sampling, arbitrary inpainting, multichannel denoising, and sequential generation **simultaneously with a single model.**
> > >
> > > We will improve our contextualization by: (1) better acknowledging existing non-scalar extensions in the literature, (2) clearly distinguishing our operator-based approach from these prior works, and (3) emphasizing how our framework uniquely combines exact sampling with multitask versatility.
> > >
> > > **Regarding technical novelty in the $\eta$ to $\eta^r $derivation:** This is an excellent point. We will add discussion of how our results for quadratic rewards relate to existing scalar interpolation results. The key technical challenge in our operator-valued case is that the relationship between $\alpha,\beta$ and $\alpha_r,\beta_r$ (Eq. 20) involves solving matrix equations rather than scalar relationships.  We will expand the technical discussion to clarify this point and explain better when Eq. (20) admits a solution.

---

> > > > ### Comment · Reviewer_MR8W · 2025-08-08
> > > >
> > > > Thank the authors for their response. I've raised my rating accordingly and recommend acceptance.

---

### Official Review · Reviewer_sQN9 · 2025-07-03

**Clarity:** 4
**Significance:** 3
**Originality:** 4
**Rating:** 5
**Confidence:** 2

**Summary:**

The aim of this paper is to train a single model that can perform various tasks. To do this, the paper generalizes the interpolants used in flow matching to accommodate diverse tasks. The suggested approach introduces additional terms in the training loss for the various interpolants but after training various tasks can be performed. Theoretical justifications are provided for the generalization of the interpolants and moderate-scale experiments are provided.

**Questions:**

Could the authors provide metrics that compare the multi-tasked trained models with single tasked based models in order to clarify the trade-offs involved?

**Ethical Concerns:**

["NO or VERY MINOR ethics concerns only"]

**Limitations:**

Limitations are discussed.

**Paper Formatting Concerns:**

None.

**Quality:**

4

**Strengths And Weaknesses:**

# Strengths

The approach is novel and allows a single model to perform diverse tasks. The paper's method is well justified, and the writing is clear.

# Weaknesses

1. The main weakness of this work is the limited scale of the experiments, although it is largely sufficient as a proof-of-concept. The work would be strengthened by a discussion of the trade-offs of having such a general model: how does it compare with a bespoke model for the various tasks considered?
2. It appears that invertible maps $\alpha$ and $\beta$ are necessary for the results to hold. The work would be strengthened by having more discussions around the choices of $\alpha$ and $\beta$. I believe this introduces a limitation on the diversity of tasks the approach can handle.

---

> ### Author Rebuttal · Authors · 2025-07-30
>
> Thank you for the positive evaluation and acceptance recommendation. We want to emphasize that our work represents a **fundamental conceptual breakthrough** - moving beyond traditional time-based diffusion to operator-based interpolation, with **theoretically guaranteed unbiased sampling** unlike existing approximation-based methods.
>
> >***Larger-scale experiments:***
>
>  Since submission, we have conducted extensive experiments on CelebA (128×128) and FFHQ (256×256) datasets to demonstrate scalability and  benchmark our approach against different methods in various image restoration tasks, evaluating the average Peak Signal-to-Noise Ratio (PSNR) and Structural Similarity Index (SSIM) on 100 test images from each dataset. To assess the performance of our methodology using Hadamard product-based operators, we employed two types of masking: square masks of sizes $40\times40$ and $80\times80$ with added Gaussian noise of standard deviation 0.05, and random masks covering $70\%$ of image pixels with Gaussian noise of standard deviation 0.01.
>
> The results reported in the table below show that our method consistently ranks either first or second in both reconstruction metrics across all tasks and datasets. Regarding visual quality, our method generates realistic, artifact-free images.
>
> **CelebA Dataset (128×128)**
> | Method     | Random PSNR | Random SSIM | Block PSNR | Block SSIM |
> |------------|-------------|-------------|------------|------------|
> | Degraded   | 13.35       | 0.234       | 21.50      | 0.744      |
> | OT-ODE [1]     | 28.84       | 0.838       | 23.88      | 0.874      |
> | D-Flow [2]    | 31.37       | 0.888       | 26.69      | 0.833      |
> | Flow-Priors [3]| 31.76       | 0.909       | 25.85      | 0.822      |
> | PnP-Flow [4]  | 32.98       | 0.930       | 26.87      | 0.904      |
> | Ours       | **33.11**   | **0.945**   | **26.96**  | **0.914**  |
>
> **AFHQ-Cat Dataset (256×256)**
> | Method     | Random PSNR | Random SSIM | Block PSNR | Block SSIM |
> |------------|-------------|-------------|------------|------------|
> | Degraded   | 11.82       | 0.197       | 22.12      | 0.742      |
> | OT-ODE [1]    | 28.36       | 0.865       | 28.84      | 0.914      |
> | D-Flow [2]    | 33.07       | 0.938       | 29.70      | 0.893      |
> | Flow-Priors [3]| 32.33       | 0.945       | 29.40      | 0.858      |
> | PnP-Flow [4]  | 33.54       | 0.953       | **30.59**  | **0.943**  |
> | Ours       | **33.76**   | **0.967**   | 29.98      | 0.938      |
>
>
> These experiments indicate that our operator-based interpolants maintain their versatility at scale, and confirm that our approach scales effectively while preserving the key advantage of training once and adapting at inference time.
>
> In addition, we have preliminary results on a **robotics maze planning experiment**, as is done in [5]. In this setting, given starting and terminal coordinates in a maze, the aim is to construct a path between the points along the maze without hitting walls or obstacles. Our method allows the zero-shot generation of this path for any pair of points in the maze without re-training. In our preliminary results, we see a 100% success rate in this task, and are also in direct communication with the authors of [5] to produce their reward metrics, if that is of interest to the reviewers.
>
> >***Trade-off analysis***:
>
> These quantitative results directly address your question about trade-offs. While our multitask model has  higher computational overhead during training, it achieves competitive or superior performance compared to specialized methods across tasks while eliminating the need for separate models. **Critically, unlike existing multitask approaches that introduce systematic biases through approximations, our method provides exact, unbiased sampling for all tasks.**
>
> >***Invertible operators $\alpha$ and $\beta$:***
>
> We appreciate this question, but note that invertibility is not required for all tasks - it primarily helps with computational efficiency by allowing us to train one η field and deduce the other from relation (5), rather than having to train both η₀ and η₁ separately. For our main applications (Hadamard products, Fourier domain operations), invertibility is naturally satisfied.
>
>
> **References:**
>
> [1] A. Pokle, M. Muckley, R. Chen, B. Karrer, "Training-free linear image inverses via flows" (TMLR)
>
> [2] H. Ben-Hamu, O. Puny, I. Gat, B. Karrer, U. Singer, Y. Lipman, "D-Flow: Differentiating through Flows for Controlled Generation"
>
> [3] Y. Zhang, P. Yu, Y. Zhu, Y. Chang, F. Gao, Y. Nian Wu, O. Leong, "Flow Priors for Linear Inverse Problems via Iterative Corrupted Trajectory Matching" (Neurips 2024)
>
> [4] S. Martin, A. Gagneux, P. Hagemann, G. Steidl, "PnP-Flow: Plug-and-Play Image Restoration with Flow Matching" (ICLR 2025)
>
> [5] B. Chen, D. Marti Monso, Y. Du, M. Simchowitz, R. Tedrake, V. Sitzmann "Diffusion Forcing: Next-token Prediction Meets Full-Sequence Diffusion" (NeurIPS 2024)

---

### Note · Authors · 2025-08-12

We thank the reviewers for their constructive feedback and the opportunity to clarify our contributions.

All reviewers acknowledged the **novelty and theoretical soundness of our approach.** Our work introduces a fundamental paradigmatic shift from time-based diffusion to operator-space traversal for multitask generation. This is not an incremental improvement but a complete reconceptualization that enables **theoretically exact, unbiased sampling across multiple tasks with a single model** that can simultaneously handle inpainting, sequential generation, multichannel denoising, and posterior sampling - a feat that no other existing method can achieve.

The reviewers also expressed some reservations about the scale of our numerical experiments. To **address their numerical concerns,** we provided extensive new numerical results demonstrating **single model versatility and scalability**:
- **CelebA (3×128×128, 49K pixels) and AFHQ-Cat (3×256×256, 196K pixels)** with state-of-the-art performance across multiple tasks and **comprehensive benchmarking** against 5 specialized methods, consistently ranking first or second across all metrics - we stress that these experiments were conducted in pixel rather than latent space, making them large-scale;
- **100% success rate** on robotics maze planning, demonstrating broad applicability beyond image tasks.

These results indicate **strong scalability potential** while maintaining the **key theoretical advantage**: training once and adapting to diverse tasks with exact guarantees. The combination of theoretical rigor (acknowledged by all reviewers) and demonstrated experimental scalability establishes a new foundation for universal generative models.

In view of the reviewers consensus on theoretical novelty and our comprehensive numerical validation, we trust that you will find our paper represents a valuable contribution to the field.

---

### Decision · Program_Chairs · 2025-09-17

**Decision:**

Accept (spotlight)

**Comment:**

This manuscript proposed a framework of operator-based interpolants, that aims to learn maps between probability distributions instead of flows as in existing flow-based generative models. Such framework enables construction of versatile models capable of multi tasks. The approaches proposed in this manuscript are sound in theory and also achieving nice empirical results in comprehensive numerical validation. The reviewers unanimously recommend acceptance after the discussion period. Thus the meta-reviewer recommends acceptance of the manuscript.